# A Novel Benchmark for Few-Shot Semantic Segmentation in the Era of Foundation Models

**Reda Bensaid**                                             *reda.bensaid@imt-atlantique.fr*
*IMT Atlantique, Brest, France*
*Polytechnique Montréal, Canada*

**Vincent Gripon**                                          *vincent.gripon@imt-atlantique.fr*
*IMT Atlantique, Brest, France*

**François Leduc-Primeau**                              *francois.leduc-primeau@polymtl.ca*
*Polytechnique Montréal, Canada*

**Lukas Mauch**                                                 *Lukas.Mauch@sony.com*
*Sony Europe, B.V. Stuttgart Laboratory 1, Germany*

**Ghouthi Boukli Hacene**                              *Ghouthi.BoukliHacene@sony.com*
*Sony Europe, B.V. Stuttgart Laboratory 1, Germany*
*Mila, Montreal, Canada*

**Fabien Cardinaux**                                         *Fabien.Cardinaux@sony.com*
*Sony Europe, B.V. Stuttgart Laboratory 1, Germany*

**Reviewed on OpenReview:** *https://openreview.net/forum?id=5EXrH2h3I5*

## Abstract

Few-shot semantic segmentation (FSS) is a crucial challenge in computer vision, driving extensive research into a diverse range of methods, from advanced meta-learning techniques to simple transfer learning baselines. With the emergence of vision foundation models (VFM) serving as generalist feature extractors, we seek to explore the adaptation of these models for FSS. While current FSS benchmarks focus on adapting pre-trained models to new tasks with few images, they emphasize in-domain generalization, making them less suitable for VFM trained on large-scale web datasets. To address this, we propose a novel realistic benchmark with a simple and straightforward adaptation process tailored for this task. Using this benchmark, we conduct a comprehensive comparative analysis of prominent VFM and semantic segmentation models. To evaluate their effectiveness, we leverage various adaption methods, ranging from linear probing to parameter efficient fine-tuning (PEFT) and full fine-tuning. Our findings show that models designed for segmentation can be outperformed by self-supervised (SSL) models. On the other hand, while PEFT methods yields competitive performance, they provide little discrepancy in the obtained results compared to other methods, highlighting the critical role of the feature extractor in determining results. To our knowledge, this is the first study on the adaptation of VFM for FSS. [1].

## 1 Introduction

Semantic segmentation, the task of pixel-level object classification within images, plays a pivotal role in computer vision applications (Minaee et al., 2022). This precise level of image understanding facilitates numerous applications across various domains, such as autonomous driving (Ha et al., 2017), seismic imagery analysis (Civitarese et al., 2019), aerial imagery (Tang et al., 2023) and medical imaging (Petitjean et al., 2015; Wang et al., 2019) where precise image analysis is crucial.

---

[1]Code to be released at: **https://github.com/RedaBensaidDS/Foundation_FewShot**

In recent years, few-shot semantic segmentation (FSS) (Catalano & Matteucci, 2023) has emerged as a challenging yet crucial area of research (Shaban et al., 2017). This paradigm seeks to train or adapt models to previously unseen object classes with minimal labeled data. The typical benchmarks for FSS involve datasets split into four folds with different classes: three are used for training or creating training episodes, while the fourth fold is reserved for testing or generating few-shot tasks, all containing an equal number of classes (Shaban et al., 2017; Nguyen & Todorovic, 2019).

Recently, Vision Foundation Models (VFM) have emerged as a pivotal advancement in computer vision. These models are extensively pre-trained on vast amounts of data and then fine-tuned for specific downstream tasks. Unlike traditional models trained from scratch on task-specific data, VFM learn rich, general-purpose representations, which can be transferred to a wide range of tasks and domains. The key advantage of VFM lies in their ability to capture complex patterns and semantic relationships present in diverse datasets, making them highly effective for tasks requiring cross-domain generalization. This has led us to explore the potential of adapting VFMs for FSS.

Although PASCAL-$5^i$ and COCO-$20^i$ serve as standard benchmarks for FSS (Catalano & Matteucci, 2023), they exhibit several limitations and do not align well with the capabilities of VFMs, which, we argue, allow for more general, efficient, and realistic use cases. More specifically, current benchmarks have the following limitations:

- They focus mainly on in-domain generalization, where models are trained on specific classes and then evaluated on few-shot tasks from the remaining classes within the *same* dataset. VFMs, by contrast, can generalize to new data without the need to observe distinct yet similar distributions from the same dataset beforehand. As such, VFMs can be deployed in more diverse conditions.

- They assume that the classes of interest were not seen during pretraining of the VFM, which is unverifiable when it comes to foundation models, and also a restrictive setting when it comes to actual applications.

- They only employ binary masks during training phases, where a single class is annotated as foreground and all other pixels are treated as background in a support image[2]. This approach, adapted from few-shot classification, enables controlled construction of few-shot episodes by ensuring a fixed number of samples per class. However, it introduces inconsistencies, such as the same class being labeled as foreground in one episode and background in another, which can hinder learning (Yang et al., 2021). Additionally, it limits the use of multi-class context during training, which is often present in realistic scenes.[3]

- They exhibit balanced class distributions when constructing few-shot tasks. This design simplifies performance evaluation, but does not reflect the natural statistics of real-world semantic segmentation datasets, where certain classes occur far more frequently than others. As a result, benchmarks with artificially balanced class sampling may misrepresent the challenges models face in realistic deployment scenarios.

In this paper, we introduce a new realistic benchmark meant to improve the following aspects:

- Our benchmark does not rely on training or fine-tuning on a subset of classes before evaluating on the rest. Instead, we directly perform few-shot adaptation on the target task, without prior exposure to the dataset. This design better reflects real-world deployment scenarios of VFMs, which are expected to generalize to novel tasks with minimal supervision and without task-specific pretraining.

- A stronger emphasis on adaptation rather than requiring the model to generalize to unseen classes in few-shot tasks [4], which aligns better with the goal of VFMs (Bommasani et al., 2021, Section 4.3).

---

[2]While existing FSS benchmarks do support multi-class segmentation in the final query images, they often rely on binary masks during the training phase for each class in the N-way K-shot setting.

[3]The recent Generalized Few-shot Semantic Segmentation (GFSS) benchmark (Tian et al., 2022) supports multi-class segmentation. However, this benchmark relies on the existence of base classes, which is incompatible with our benchmark as stated above.

[4]Recent benchmarks have started to address cross-domain few-shot segmentation. However, these approaches often involve training or meta-training on one dataset and then testing on another, which still differs from our objective.

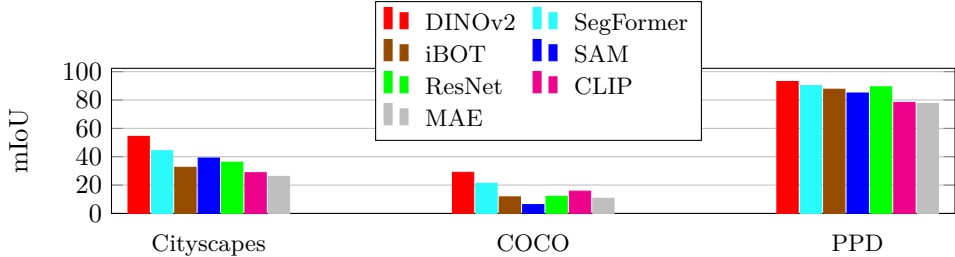

Figure 1: mIoU of various pretrained models when adapted to 3 different 1-shot segmentation tasks (only best performance achieved out of 5 tested methods is displayed).

- Our approach uses fully labeled few-shot examples, allowing for multi-class supervision. This setup enables richer contextual understanding, avoids semantic inconsistencies during training, and supports more complex few-shot scenarios. It is also compatible with standard semantic segmentation datasets and aligns with application domains, such as autonomous driving and remote sensing—where full-scene understanding is typically required (Çağrı Kaymak & Uçar, 2018; Peng et al., 2022; Holder & Shafique, 2022).

- Class balance is not strictly enforced, instead, our benchmark reflects the natural class distribution present in the selected datasets. For instance, in datasets like Cityscapes, frequently occurring classes such as "car" or "road" appear in most images, while other classes are comparatively rare. Preserving this imbalance provides a more realistic evaluation of model performance, particularly in terms of handling dominant and rare classes under practical conditions.

The primary objective of our research is to investigate which vision foundation model and adaptation method yields the most effective few-shot semantic segmentation pipeline. We are the first, to our knowledge, to systematically test the adaptation capabilities of renowned foundation models to this specific challenge. To this end, we have introduced a novel benchmark, constructed using three widely-recognized semantic segmentation datasets, and systematically evaluated multiple adaptation methods. The five foundation models under consideration are DINOv2 (Oquab et al., 2023), Segment Anything (SAM) (Kirillov et al., 2023), CLIP (Radford et al., 2021), Masked AutoEncoder (MAE) (He et al., 2021) and iBOT (Zhou et al., 2022a). Additionally, we have carried out extensive experiments to better understand the elementary contribution of foundation models design (e.g. type of architecture, size, training dataset) and the effect of the adaptation methods onto the obtained performance.

Our findings indicate that, surprisingly, models designed for segmentation are consistently outperformed by a self-supervised learning (SSL) model, DINOv2, across various datasets and adaptation methods. While Segment Anything (SAM) generally provides competitive results, it underperforms on some datasets (Figure 1). Additionally, our study indicates that parameter-efficient fine-tuning (PEFT) methods yield highly competitive performance, although the performance differential with other methods is not substantial. This underscores the critical role of feature extractors in model performance. We anticipate that our research will assist in selecting appropriate solutions for few-shot semantic segmentation tasks and contribute to the rigorous benchmarking of this evolving field.

## 2 Related Work

In this section, we discuss related work in the literature in few-shot classification and FSS.

### 2.1 Few-Shot Classification

Few-shot classification initially focused on in-domain tasks, with benchmarks involving dataset splitting into training, validation, and fake few-shot task generation for evaluation (Ravi & Larochelle, 2017; Snell et al., 2017; Vinyals et al., 2016; Finn et al., 2017). The introduction of Meta-dataset (Triantafillou et al., 2020) marked a shift towards the cross-domain setting, where the dataset used for model training differs from the tasks evaluated later, aligning with more realistic scenarios. More recently, the field has embraced the use of foundation models, adapting them to few-shot tasks. Notably, benchmarks such as those in (Zhou et al., 2022b) explore the application of CLIP (Radford et al., 2021) to 11 downstream tasks, and DINO(Caron et al., 2021) to few-shot downstream tasks(Luo et al., 2023).

Methodologically, diverse approaches have emerged, ranging from hallucination-based (Li et al., 2020; Hariharan & Girshick, 2017) to meta-learning (Finn et al., 2017; Munkhdalai et al., 2018; Zhang et al., 2021; Munkhdalai & Yu, 2017). Straightforward methods have also demonstrated effectiveness (Li et al., 2021; Tian et al., 2020; Li et al., 2022). Recent surveys, such as (Luo et al., 2023), suggest that optimal approaches often involve simple finetuning atop competitive pretrained models, augmented with an additional linear layer. This observation motivates the methods introduced in our paper.

## 2.2 Semantic Segmentation and Few-Shot Semantic Segmentation

Few-shot semantic segmentation has been an active area of research for several years. Early works, such as (Shaban et al., 2017) and (Nguyen & Todorovic, 2019), introduced benchmarks like PASCAL-$5^i$ and COCO-$20^i$, which share similarities with the in-domain benchmarks commonly used in few-shot classification.

Subsequent studies leveraged pretrained models on ImageNet (Yang et al., 2021; Hong et al., 2022; Wu et al., 2021; Okazawa, 2022), drawing parallels to the Meta-dataset approach in few-shot classification. Despite this progression, these methods continued to rely on the earlier benchmarks, limiting their applicability to broader scenarios.

More recently, efforts have shifted toward cross-domain few-shot semantic segmentation (Wang et al., 2022; Nie et al., 2024; Lei et al., 2022). These approaches often depend on meta-training and assume that images from the target domain originate from a distribution distinct from the training dataset. However, this assumption may not hold in scenarios involving foundation models, where domain boundaries are less clearly defined.

Recent works focused on adapting vision foundation models to FSS (Shuai et al., 2023; Liu et al., 2024; Sun et al., 2024; Meng et al., 2024; Zhu et al., 2024; Zhang et al., 2024), However, these studies mainly explore specific model adaptations, without offering a broad comparative analysis of their effectiveness as feature extractors. Our work provides a benchmark and a comprehensive comparative analysis of these models alongside various adaptation methods, making our contribution orthogonal to existing research.

Additionally, many existing methods focus on binary labeling for the few-shot setting, where only a single object of interest is annotated in the support images, and all other objects are treated as background. This simplification can hinder effective training, particularly in tasks requiring more nuanced multi-class segmentation.

## 3 Benchmark

### 3.1 Task Formulation

In accordance with the established terminology of few-shot classification (Snell et al., 2017), our benchmark revolves around the concept of a support set $\mathcal{S}$ and a query set $\mathcal{Q}$. The support set $\mathcal{S}$ comprises a limited number of images, each accompanied by their corresponding, fully labeled, ground truth segmentation. These support images are employed for the *training* or *calibration* of a generic semantic segmentation model. In contrast, the query set $\mathcal{Q}$ serves as the evaluation dataset on which we compute the mean Intersection-over-Union (mIoU). In practical applications, $\mathcal{Q}$ is typically not available and is utilized exclusively for benchmarking purposes.

To compute the mIoU, we calculate the Intersection-over-Union (IoU) for each class $i$ and then average these values:

$$\text{mIoU} = \frac{1}{n} \sum_{i=1}^{n} \frac{\text{TP}_i}{\text{TP}_i + \text{FP}_i + \text{FN}_i},$$

where $n$ represents the total number of classes or objects, $\text{TP}_i$ is the true positive count for class $i$, $\text{FP}_i$ is the false positive count for class $i$, and $\text{FN}_i$ is the false negative count for class $i$. One noteworthy distinction from few-shot classification is that individual images typically contain instances from various classes of interest. As such, it is challenging to define tasks in the $k$-shot manner, where each class is associated with exactly $k$ elements in the support set.

In our proposed framework, we introduce a more practical approach. We define a $k$-shot segmentation task as one in which the support set $\mathcal{S}$ contains *at least* $k$ instances of each class. Using our proposed

sampling protocol described in the next subsection, in a scenario involving a total of $n$ classes, a $k$-shot task mandates the inclusion of precisely $nk$ unique images within the support set.

### 3.2 Benchmark Generation

To construct few-shot tasks from readily available semantic segmentation datasets, we employ the following sampling procedure:

1. Initialize an empty support set, denoted as $\mathcal{S} = \emptyset$.

2. For each class $i$ (excl. the background class) within the chosen dataset $\mathcal{D}$, we compile a list of all training images that contain at least one instance of that class, resulting in $\mathcal{D}_i = \{\text{image} \in \mathcal{D} \mid \text{image contains instances of class } i\}$.

3. Following an arbitrary order of classes, we randomly select $k$ images from $\mathcal{D}_i$ without replacement to include in the support set $\mathcal{S}$. We ensure that images are only selected once.

4. If there are not enough remaining images for a specific class $i$, we return to Step 1. It is important to note that this reset condition did not arise during our experiments.

## 4 Datasets

In constructing our proposed benchmark, we have selected three prominent semantic segmentation datasets, each offering unique characteristics:

**Cityscapes (Cordts et al., 2016; 2015)** is a large scale dataset on semantic understanding of urban scene streets. It contains 2,975 images for training, 500 for validation and 1,525 for testing. It is annotated using 19 classes (such as "car", "road", "building") and has input resolution 1024x2048.

**COCO (Microsoft Common Objects in Context) (Lin et al., 2014)** is a large scale dataset used for multiple tasks including semantic segmentation. The 2017 release contains 118k images for training, 5k for validation and 41k for testing. It is annotated using 80 classes (such as "person", "bicycle", "elephant"). Images in COCO often exhibit varying input resolutions, commonly falling between 400 and 640 pixels for both width and height.

**PPD (PLANT PHENOTYPING DATASETS) (Minervini et al., 2016; Hanno Scharr; Sotirios A. Tsaftaris)** is a small dataset used for multiple tasks including plant segmentation (foreground to background) and leaf segmentation (multi-instance segmentation). We focus in our work on the plant segmentation task, where the dataset contains 783 images and 2 classes (foreground-background segmentation). Typical input resolutions for PPD images hover around 500x500. Contrary to the two other datasets, we considered the background to be a class of its own for benchmark generation, meaning that a 1-shot problem would contain 2 support images.

The inclusion of these three datasets ensures a diverse evaluation framework, spanning different levels of difficulty and data distributions. From simpler binary segmentation tasks, such as foreground-background segmentation in PPD, to complex multi-class problems like COCO with 80 classes, this benchmark enables a comprehensive assessment of model performance across varied scenarios. Foreground-background segmentation, exemplified by PPD, holds significant relevance in domains like medical imaging (Dumitru & Peteleaza, 2023) and road extraction (Aich et al., 2018). Despite its smaller scale, PPD demonstrates the importance of addressing practical, real-world segmentation challenges that demand robust and efficient solutions. This approach aligns with practices in few-shot classification, such as the Coop benchmark, which utilized 11 publicly available datasets (Zhou et al., 2022b).

## 5 Backbones and Adaptation Methods

In this section, we introduce the methodologies that form the basis of our comparative study. Drawing from existing literature on both few-shot classification (Luo et al., 2023) and few-shot semantic segmentation (Catalano & Matteucci, 2023), we have identified that the majority of methods rely on a combination of two fundamental components.

The first essential component is a *pretrained model*, typically a transformer (Vaswani et al., 2017) or a convnet. It's important to note that while these models are not specifically designed for semantic

segmentation, they bring a substantial amount of valuable knowledge that can be leveraged in this context.

The second critical component is an *adaptation method*, a mechanism devised to tailor the knowledge encoded within the pretrained model for effective utilization in the specific task of interest.

Next, we provide comprehensive details on the specific models and adaptation methods that constitute the experimental foundation of our study.

## 5.1 Pretrained Models

In our study, we investigate five pretrained models, each offering unique characteristics and advantages (see also summary of the models and their main components in Table 10):

**MAE (Masked AutoEncoder) (He et al., 2021)** is a model pretrained on ImageNet-1K (Deng et al., 2009) using a self-supervised technique involving masking portions of input images and reconstructing them using partial tokens from the unmasked segments. It comprises a Vision Transformer (ViT) (Dosovitskiy et al., 2021) component that maps observed image regions into latent representations and a decoder for image reconstruction. Previous research (He et al., 2021) has demonstrated MAE's effectiveness in transfer learning for segmentation tasks, particularly in semantic segmentation.

**SAM (Segment Anything Model) (Kirillov et al., 2023)** is a recent foundation model for segmentation, pretrained on the SA-1B dataset, which includes 11 million images and 1 billion segmentation masks. SAM exhibits robust generalization capabilities across a broad range of segmentation tasks. It utilizes a Vision Transformer (ViT) (Dosovitskiy et al., 2021) initialized with a MAE (He et al., 2021), a prompt encoder, and a mask decoder. SAM's image encoder generates embeddings from input images, which are subsequently employed by the mask decoder in conjunction with the prompt encoder. While SAM's mask decoder outputs class-agnostic masks intended for segmenting instances or objects at various granularities, our specific task focuses exclusively on the encoder's role as a feature extractor and does not require prompts.

**iBOT (Zhou et al., 2022a)** is a self-supervised learning model based on both masked image modeling and knowledge distillation. It comprises a ViT trained either on ImageNet-1K (Deng et al., 2009)or on ImageNet-22k and has show great generalization capabilities on multiple downstream tasks. In our study, we use the ViT trained on Imagenet-1K since it has better downstream tasks performance in (Zhou et al., 2022a).

**DINOv2 (Oquab et al., 2023)** is a foundational vision model trained using the self-supervised technique (Caron et al., 2021) called DINO and iBOT (Zhou et al., 2022a). This methods consists in having two networks with the same architecture, namely the teacher and the student. The student parameters are learned by minimizing a loss measuring the discrepancy between teacher and student outputs learning a "local to global" correspondence where the student processes global and local views of the image while the teacher only processes the global views. The teacher is built with an exponential moving average of past iteration of the student. The model comprises a ViT trained on a diverse range of datasets, including notable semantic segmentation datasets like Cityscapes (Cordts et al., 2016), ADE20k (Zhou et al., 2017; 2019), and Pascal VOC 2012 (Everingham et al.). DINOv2 has previously been evaluated on (not few-shot) semantic segmentation datasets in (Oquab et al., 2023), utilizing the frozen backbone and a simple segmentation decoder.

**CLIP (Contrastive Language-Image Pre-training) (Radford et al., 2021)** is a model that consists of a vision encoder and a text encoder. These encoders are jointly trained using a contrastive loss to align embeddings of images with their corresponding captions. The model is pretrained on a private dataset of 400 million (image, text) pairs and has gained significant attention for its utility in zero-shot classification on downstream tasks. In our study, we exclusively leverage the ViT version of the visual encoder.

**FCN (Fully Convolutional Network) (Long et al., 2015)** is a conventional convolutional architecture extensively used for semantic segmentation tasks. It consists of an encoder and a decoder, both of which are composed of convolutional layers. In our configuration, the encoder is based on ResNet50 (He et al., 2016) and is pretrained on a subset of COCO, specifically using the 20 classes shared with the Pascal VOC dataset. We incorporate FCN as a convolution-only benchmark for comparison.

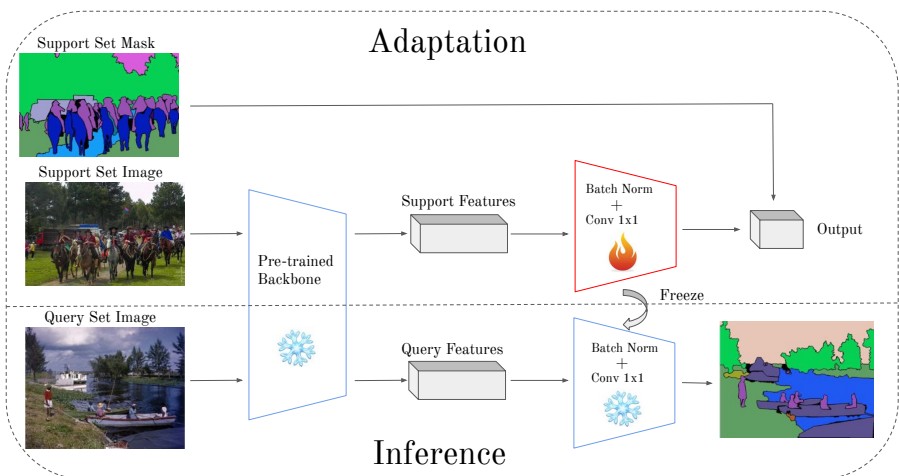

Figure 2: The pipeline of the Linear method: we freeze the backbone and train the shallow decoder on the support set, and then we test the encoder+decoder on the query set.

**SegFormer (Xie et al., 2021)** is a semantic segmentation model based on a hierarchical Transformer (Vaswani et al., 2017) encoder paired with a simple, lightweight MLP decoder. In our study, we use a SegFormer trained on ADE20K (Zhou et al., 2017) dataset, specifically utilizing only the ViT encoder. This model was selected for its recent advancements in semantic segmentation, serving as a benchmark for comparison in our study.

## 5.2   Methods

We then consider five adaptation methods in our study, ranging from linear probing methods to partial fine-tuning and full fine-tuning methods [5]:

**Linear:** A common approach (Oquab et al., 2023; Strudel et al., 2021) consists in freezing the pretrained backbone and adding a segmentation head, called *decoder*, consisting of a batch normalization layer followed by a convolutional 1x1 layer. Figure 2 illustrates the overall pipeline for this linear adaptation method.

**Multilayer:** It has been shown in (Oquab et al., 2023) and (Xie et al., 2021) that using a concatenation of multiple layers from the pretrained model can boost performance in downstream tasks. We thus propose to use a linear method on the concatenation of the last 4 blocks of the pretrained model.

**Singular Value Fine-tuning (Sun et al., 2022) (SVF):** The SVF method decomposes parameters into three matrices using Singular Value Decomposition (SVD) and fine-tunes only the singular values diagonal matrix while keeping the rest frozen. In (Sun et al., 2022), this approach demonstrated promising results in few-shot semantic segmentation tasks, showcasing its ability to specialize the pretrained model without overfitting. In our study, we initially apply the linear method and then proceed to fine-tune the singular values of the encoder alongside the decoder parameters.

**Low-Rank Adaptation (Hu et al., 2022) (LoRA):** In this approach, we maintain the pretrained backbone's frozen state and introduce low-rank trainable adapters, as described in (Hu et al., 2022). Similarly to the SVF method, LoRA enables tuning only a limited number of parameters. Our procedure begins with training the decoder using the linear method and then introduces LoRA adapters to the encoder. Subsequently, we fine-tune both the adapters and decoder parameters.

**Fine-tuning:** The fine-tuning method, a straightforward yet effective approach, involves fine-tuning the entire pretrained model. This method has demonstrated state-of-the-art performance in few-shot classification (Luo et al., 2023). However, it is susceptible to overfitting due to the limited number of training samples compared to the typically extensive parameter pool in the considered encoders.

---

[5]We do not consider meta-learning methods in our study, as these approaches rely on base classes to facilitate rapid adaptation to new classes, such as in episodic training. In contrast, our benchmark directly begins training on the support set.

## 6    Experiments

### 6.1    Training Procedure

To ensure the reproducibility of our results, we conducted all experiments with three fixed random seeds and computed the mean intersection-over-union (mIoU) across these runs. For the results averaged across the datasets, we averaged the results per run corresponding to a seed and thus having the same sampled support set, and then averaged these three means to compute their mean and standard deviation. For preprocessing, we followed the approach outlined in (Gao, 2023). This involved random horizontal flipping, random scaling of the shorter side within the range of [400, 1600] while preserving the aspect ratio, and random cropping to a size of 1024x1024. We applied a subset of RandAug (Cubuk et al., 2020) operations, including auto contrast, equalize, rotate, color, contrast, brightness, and sharpness. Subsequently, all images were resized to 1024x1024 resolution, except for CLIP and MAE, which accept input sizes of 224x224. We acknowledge that the resolution variation, depending on the considered model, may introduce some analysis challenges but was an unavoidable aspect of our setup. For the learning rate, we employed the Polynomial learning rate scheduler with a power of 0.9 for the linear and multilayer methods. Specifically, we used a learning rate of 0.2 for Cityscapes, 0.05 for COCO, and 0.001 for PPD. For fine-tuning methods (SVF, LoRA, and fine-tune), we did a grid search across models and datasets with values including $10^{-2}$, $10^{-3}$, $10^{-4}$, $10^{-5}$, and $10^{-6}$ to determine the one yielding the best average performance for 1-shot. Once identified, we applied this optimal learning rate to experiments involving more shots, primarily due to computational constraints. Linear and multilayer methods were trained with a batch size of 4. For the Fine-tuning, SVF, and LoRA methods, the training phase consists of two stages. First, we train the decoder using the linear method. Then, we perform fine-tuning on both the decoder and the adapters (or all parameters in the case of fine-tuning) with a reduced batch size of 2 to address memory constraints. Training duration extended to 200 epochs for Cityscapes and PPD, and 100 epochs for COCO. The query set consisted of the standard validation sets for Cityscapes and COCO. For PPD, given the absence of a conventional split, we performed a fixed 50/50 random split into training and validation sets. Our models were trained on a combination of NVIDIA A6000, NVIDIA RTX3090 and NVIDIA A100 GPUs. Runtime varied across datasets and models, ranging from approximately 30 minutes per run for MAE, CLIP, and ResNet on Cityscapes to approximately 4 hours per run for DINOv2 on COCO. For the size of the models, we chose the base version of the vision transformers, leading to a 89M parameters model for SAM and 86M for DINOv2, CLIP and MAE. For the FCN we chose a ResNet50 as an encoder, leading to a 23M parameters model.

### 6.2    What Works The Best?

First, we aim to identify the optimal combination of pretrained models and adaptation methods based on mIoU. Table 1 presents the average performance on the three considered datasets for all combinations of pretrained models and adaptation methods, along with their standard deviation over 3 runs (6.1), in the context of $\{1, 2, 5, 10\}$-shots. Several conclusions can be drawn:

- DINOv2 consistently outperforms all other models across various settings. This superiority is especially notable on Cityscapes and COCO, where DINOv2 surpasses other models by a significant margin (see Appendix Tables 3, 4, 5, 6 for the detailed results). Surprisingly, it outperforms supervised models pre-trained on other semantic segmentation datasets, showing the potential superiority of VFM adaptation over cross-domain generalization. However, since Cityscapes is included in DINOv2's pretraining data, albeit without segmentation masks, some degree of data leakage may be present. As stated in the introduction, our primary objective is to assess the adaptation capabilities of off-the-shelf VFM rather than their generalization abilities. Recent studies (Udandarao et al., 2024) have begun exploring this issue, emphasizing the need for further research into the influence of pretraining data distribution on downstream tasks.

- SAM encoder exhibits strong performance on average. It outperforms CLIP, MAE, and ResNet on both Cityscapes and PPD. However, it yields comparatively poor results on COCO (refer to Appendix Tables 3, 4, 5, 6). This difference is likely attributed to our decision to retain the SAM image encoder while discarding the original mask decoder A.3.

- The multilayer method outperforms the linear approach for most models, indicating that including earlier feature maps, which are more locally focused (Raghu et al., 2021; Dosovitskiy

Table 1: mIoU averaged over our three considered datasets in $k$-shot semantic segmentation for $k \in \{1, 2, 5, 10\}$. Bold numbers correspond to the best performance across extractors and underline numbers to the best performance across adaptation methods.

|  |  | Linear | Multilayer | SVF | LoRA | Fine-tuning |
|---|---|---|---|---|---|---|
| **1-shot** | SAM | 40.66±09.00 | 43.36±02.52 | 41.84±16.83 | 43.10±03.38 | 42.35±09.91 |
|  | DINOv2 | **54.78**±03.09 | **53.51**±03.10 | **57.77**±04.20 | **57.67**±02.50 | **57.21**±04.04 |
|  | iBOT | 41.28±02.87 | 41.24±01.80 | 43.04±03.31 | 43.10±02.67 | 43.70±02.91 |
|  | CLIP | 35.88±10.90 | 39.57±07.29 | 40.90±13.93 | 38.23±04.50 | 38.69±10.20 |
|  | MAE | 35.50±10.16 | 36.65±07.38 | 37.76±08.08 | 36.37±04.22 | 36.46±10.28 |
|  | FCN-ResNet50 | 37.78±04.17 | 44.08±02.16 | 43.40±08.63 | 41.45±03.70 | 39.63±06.49 |
|  | SegFormer | 45.60±02.40 | 48.96±01.90 | 50.92±02.41 | 49.93±01.23 | 51.16±01.83 |
| **2-shot** | SAM | 42.30±03.72 | 46.78±01.61 | 46.56±01.96 | 45.78±02.94 | 45.73±02.16 |
|  | DINOv2 | **61.20**±00.29 | **59.21**±00.83 | **64.10**±00.83 | **63.73**±01.28 | **64.35**±00.33 |
|  | iBOT | 45.60±01.75 | 46.20±01.02 | 47.35±02.20 | 47.81±02.54 | 47.91±01.55 |
|  | CLIP | 39.08±03.45 | 43.46±00.70 | 45.59±01.94 | 42.72±02.97 | 35.68±07.11 |
|  | MAE | 36.90±04.15 | 38.73±02.39 | 40.36±03.83 | 40.37±04.27 | 39.15±03.50 |
|  | FCN-ResNet50 | 40.98±01.96 | 47.36±00.85 | 48.66±00.92 | 46.55±02.36 | 48.81±00.32 |
|  | SegFormer | 50.79±01.32 | 55.58±01.92 | 54.62±01.85 | 52.87±01.52 | 55.48±00.29 |
| **5-shot** | SAM | 46.55±00.82 | 51.73±00.21 | 49.93±00.21 | 51.57±00.24 | 52.24±00.29 |
|  | DINOv2 | **66.47**±00.60 | **65.89**±00.63 | **68.64**±00.88 | **67.54**±00.20 | **68.82**±01.40 |
|  | iBOT | 50.82±00.38 | 51.08±00.16 | 53.12±00.16 | 54.67±00.04 | 54.77±00.29 |
|  | CLIP | 46.30±00.29 | 48.67±00.61 | 51.56±00.22 | 49.00±00.39 | 43.60±00.76 |
|  | MAE | 44.36±00.23 | 45.02±00.15 | 46.20±00.92 | 47.02±00.16 | 46.68±00.34 |
|  | FCN-ResNet50 | 47.02±00.48 | 49.70±00.27 | 51.58±00.03 | 51.40±00.28 | 51.51±00.26 |
|  | SegFormer | 56.85±00.06 | 60.93±00.16 | 59.08±00.69 | 58.30±00.22 | 60.20±00.63 |
| **10-shot** | SAM | 47.79±00.39 | 54.26±00.11 | 51.41±00.29 | 54.20±00.37 | 55.20±00.48 |
|  | DINOv2 | **69.86**±00.23 | **69.95**±00.12 | **69.97**±00.69 | **69.25**±00.25 | **71.07**±01.35 |
|  | iBOT | 54.41±00.35 | 54.51±00.26 | 55.17±00.37 | 58.00±00.58 | 58.59±00.64 |
|  | CLIP | 50.14±00.41 | 52.49±00.41 | 53.64±00.77 | 51.92±00.41 | 47.62±03.36 |
|  | MAE | 47.56±00.41 | 47.97±00.28 | 46.73±00.17 | 42.94±03.07 | 48.63±00.30 |
|  | FCN-ResNet50 | 48.69±00.69 | 51.98±00.25 | 53.04±00.31 | 53.02±00.25 | 52.86±00.55 |
|  | SegFormer | 59.53±00.52 | 64.08±00.51 | 60.99±00.70 | 60.21±00.29 | 62.46±00.56 |

et al., 2021), provides the decoder with greater segmentation flexibility. Furthermore, it exhibits a lower standard deviation, indicating greater stability across the different runs of our proposed benchmark.

- Despite variations in the number of trainable parameters (less than 1% for LoRA, around 2% for SVF, and 100% for finetune), the finetuning methods (SVF, LoRA, and Full Fine-tuning) yield comparable results between them highlighting the importance of using efficient methods.

- Fine-tuning scales better than LoRA and SVF and gives better results on average with 10 shots, this is due to the fact that we suffer less from over-fitting when having more shots.

## 6.3 Individual Factors

In the next series of experiments, we study the effect of various elements that could influence the performance of the models. Namely, these include the model size, the underlying architecture, the pretrained dataset, the pretraining method and the adaptation method. To isolate the effects of each component, we compare foundation models that differ primarily in one of these aspects, although minor variations may exist between the models. By studying these individual components, we seek to provide a more granular understanding of what drives model performance. (Other individual factors such as input resolution and registers are discussed in the Appendix A.5).

### 6.3.1 Model Size.

Table 2 demonstrates that a larger model size does not necessarily translate to a big boost of performance. Specifically, for SAM and DINOv2 with the linear method, the base model (ViT-B) yields, on average, competitive performance, despite having 3 times less parameters than the large (ViT-L) models. In the case of ResNet, moving from ResNet-50 (23M parameters) to ResNet-101 (44M parameters) shows a performance boost, though the increase is modest.

For the LoRA method, fine-tuning of a subset of parameters of the encoder yields better performance with bigger models. These observations partially accounts for the performance discrepancy between SAM, DINOv2, and ResNet50, highlighting the influence of model size differences.

For tiny models like SegFormer-B0, which has 3.7M parameters, there is a noticeable drop in performance, indicating that models adapted for device deployment struggle with poor adaptation performance. Enhancing the adaptation of these models remains an area for future research.

Table 2: Effect of different components on mIoU. Bold numbers indicate the best performance among the compared models for a specific component.

| | | Linear | | | LoRA | | |
| | | Cityscapes | COCO | PPD | Cityscapes | COCO | PPD |
|---|---|---|---|---|---|---|---|
| model size | SAM (ViT-B) | **35.72±01.50** | 03.13±00.09 | 83.14±10.36 | 38.50±00.12 | 05.85±00.13 | 84.94±10.11 |
| | SAM (ViT-L) | 35.00±01.29 | **03.43±00.07** | 72.95±09.05 | 40.70±00.45 | 07.31±00.37 | 79.14±10.75 |
| | SAM (ViT-H) | 34.64±01.36 | 03.38±00.11 | **83.34±04.02** | 40.78±00.57 | **07.46±00.29** | 85.73±08.42 |
| | SegFormer-B0 | 30.97±01.88 | 08.48±00.34 | 72.35±07.02 | 32.15±00.94 | 09.70±00.41 | 87.65±01.35 |
| | SegFormer-B5 | **40.55±03.40** | **18.37±01.13** | **77.88±07.82** | **42.34±02.20** | **19.64±00.49** | **87.82±03.78** |
| | DINOv2 (Vit-S) | 41.16±02.39 | 20.62±00.38 | 87.99±05.58 | 44.75±00.71 | 22.72±00.71 | 86.55±06.08 |
| | DINOv2 (Vit-B) | 48.80±01.82 | 23.24±00.38 | **92.31±01.84** | 54.35±01.55 | 28.99±01.33 | **89.67±06.43** |
| | DINOv2 (Vit-L) | **49.36±01.61** | **23.50±00.33** | 87.87±07.30 | **54.79±00.64** | **31.54±00.81** | 87.46±07.97 |
| | FCN-ResNet50 | 32.39±01.79 | 10.08±00.32 | **70.86±05.91** | **35.74±01.41** | 11.71±00.24 | 76.91±10.29 |
| | FCN-ResNet101 | **32.97±01.76** | **10.94±00.16** | 69.70±12.98 | 29.38±07.03 | **12.23±00.57** | **81.79±11.40** |
| training dataset | DINOv1 | 29.03±01.08 | 05.92±00.39 | 82.14±10.04 | 32.53±00.97 | 06.89±00.44 | 82.99±09.35 |
| | DINOv2 | **48.80±01.82** | **23.24±00.38** | **92.31±01.84** | **54.35±01.55** | **28.99±01.33** | **89.67±06.43** |
| | CLIP | **23.83±01.51** | 11.32±00.31 | **72.49±10.78** | **27.95±01.62** | 13.62±00.78 | **73.13±11.70** |
| | OpenCLIP | 21.01±02.28 | **11.64±00.40** | 71.85±09.85 | 22.73±01.58 | **14.34±00.06** | 71.55±09.42 |
| | MAE (IN1k) | **23.63±00.89** | **08.54±00.46** | 74.34±10.88 | **26.10±00.26** | **10.13±00.42** | 72.87±12.83 |
| | MAE (IG-3B) | 21.73±01.07 | 05.76±00.16 | **75.18±09.26** | 23.59±00.21 | 07.84±00.37 | **80.05±07.44** |
| architecture | DINOv1 (ViT-S) | **26.61±00.53** | **04.86±00.23** | **81.86±08.19** | **29.90±00.95** | **05.82±00.34** | **79.70±09.03** |
| | DINOv1 (ResNet-50) | 23.51±01.40 | 03.42±00.05 | 58.98±00.61 | 26.98±01.28 | 03.78±00.04 | 77.35±10.50 |
| | CLIP (ViT-B) | **23.83±01.51** | **11.32±00.31** | **72.49±10.78** | **27.95±01.62** | **13.62±00.78** | **73.13±11.70** |
| | CLIP (ResNet-101) | 20.22±01.57 | 02.61±00.06 | 69.00±06.03 | 08.96±00.11 | 00.91±00.03 | 67.40±11.91 |
| | OpenCLIP(ViT-B) | 21.01±02.28 | **11.64±00.40** | 71.85±09.85 | 22.73±01.58 | 14.34±00.06 | 71.55±09.42 |
| | OpenCLIP(ConvNext-L) | **29.29±2.80** | 11.30±00.02 | **81.05±08.30** | **38.52±01.14** | **17.05±00.20** | **84.94±09.72** |
| training method | DINOv1 (KD) | 29.03±01.08 | 05.92±00.39 | 82.14±10.04 | 32.53±00.97 | 06.89±00.44 | 82.99±09.35 |
| | MAE (MIM) | 23.63±00.89 | 08.54±00.46 | 74.34±10.88 | 26.10±00.26 | 10.13±00.42 | 72.87±12.83 |
| | iBOT (KD + MIM) | **29.46±01.72** | **09.31±00.68** | **85.06±08.80** | **32.54±01.43** | **11.27±00.99** | **85.48±07.80** |

### 6.3.2 Training Dataset.

To evaluate the impact of training dataset characteristics on model performance, we conducted a comparative analysis of DINOv1 and DINOv2. DINOv1 was trained on the standard ImageNet-1K dataset, while DINOv2 utilized a more extensive dataset of 142 million curated and uncurated images. Additionally, it is noteworthy that the models differ in their loss functions; however, our primary focus remains on the influence of dataset size and composition.

As illustrated in Table 2, DINOv2 significantly outperforms DINOv1, particularly demonstrating superior results on the COCO dataset. This suggests that a larger training dataset may enhance performance in few-shot semantic segmentation tasks. Nonetheless, this performance boost could also be influenced by other factors differentiating DINOv1 from DINOv2.

Furthermore, we compared CLIP, trained on 400 million (image, text) pairs, with OpenCLIP (Cherti et al., 2023) trained on the LAION-5B (Schuhmann et al., 2022) dataset comprising 5 billion (image, text) pairs, and MAE trained on ImageNet-1K against another MAE variant trained on the Instagram-3B dataset (Singh et al., 2023) , which includes 3 billion images. According to Table 2, for MAE and CLIP, an increase in training dataset size does not consistently result in better performance and can sometimes lead to a performance decline on certain datasets. Overall, the benefits of larger pre-training datasets remain ambiguous, indicating that the advantages of scaling up the training data are not universally guaranteed.

### 6.3.3 Architecture.

To assess the influence of model architecture, we analyzed the performance differences between DINOv1 (ViT-S) and its ResNet-50 variant, alongside comparing the CLIP (ViT-B) version against the CLIP ResNet-50 version. It is noteworthy that for CLIP, the ViT-B model is larger than its ResNet counterpart, an aspect previously discussed in terms of model size impact. According to Table 2, ViT architectures demonstrate a clear performance advantage over their ResNet equivalents. This trend underscores the potential superiority of ViT models in adapting to various tasks. For OpenCLIP, the ConvNext-L outperforms the ViT-B, this can be explained by the fact that we used a resolution of 1120 for ConvNext as in (Yu et al., 2023), and that CLIP only uses a resolution of 224 as discussed in A.5.

### 6.3.4 Training Method.

To investigate the effect of different self-supervised learning (SSL) methods on model performance, we conducted a comparison between knowledge distillation (KD) and masked image modeling (MIM) models. Despite our initial expectations that MIM models would surpass KD models (Park et al., 2023), the comparison in Table 2 reveals that DINOv1 (KD) surpasses MAE (MIM) on the Cityscapes and PPD datasets, and MAE outperforms DINOv1 on COCO. This mixed outcome indicates that there is no definitive answer as to which SSL training approach offers the best adaptability for FSS tasks. However, iBOT, which integrates both KD and MIM techniques, shows superior performance over both models, leveraging the strengths of both approaches.

## 7 Conclusion

We have introduced a novel benchmark specifically designed to assess the adaptation of Vision Foundation Models (VFM) to Few-Shot Semantic Segmentation (FSS). Our evaluation of multiple VFMs across this new benchmark has highlighted several key areas for further exploration : DINOv2 emerges as the top-performing backbone model, consistently outperforming other foundation models. While SAM exhibits strong overall results, it presents unexpected limitations on certain datasets. Surprisingly, our experiments underscore the competitive advantages of straightforward and efficient methods such as simple linear segmentation heads, challenging the necessity of complex procedures?

This work not only presents a comparison of techniques but also a more realistic and practical benchmark. We hope this benchmark will enhance the development of few-shot semantic segmentation methods and offer a robust framework for future comparisons.

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

# A Appendix

## A.1 Detailed Results for Every Dataset

We can see in Tables 3, 4, 5, 6 the detailed results on the 3 datasets separately:

- The range of performance on the 3 datasets is different mainly due to the varying size and number of classes. On PPD, we only have 2 classes making the task easier but the training is more unstable due to the restricted number of training examples, since we only have 2 images in our support set. This leads to a bigger standard deviation in PPD compared to COCO and Cityscapes since the choice of these 2 images is crucial and the images are different between the 3 folders constituting the dataset.

- SAM image encoder performance is poor on COCO compared to other models even though it yielded good performance in the other 2 datasets. This is probably due to the fact that we isolated the image encoder (the adaptation of the whole SAM model to prompt free Few-Shot Semantic Segmentation is left to future work).

Table 3: Detailed results of mIoU on our three considered datasets in 1-shot semantic segmentation.

| | | Linear | Multilayer | SVF | LoRA | Fine-tuning |
|---|---|---|---|---|---|---|
| Cityscapes | SAM | 35.72±01.50 | 39.06±01.97 | 38.90±00.48 | 38.50±00.12 | 38.14±00.40 |
| | DINOv2 | **48.80±01.82** | **46.77±02.80** | **51.96±01.37** | **54.35±01.55** | **50.87±01.25** |
| | iBOT | 29.46±01.72 | 26.84±02.04 | 32.16±01.20 | 32.54±01.43 | 31.78±01.31 |
| | CLIP | 23.83±01.51 | 26.10±02.01 | 28.72±01.06 | 27.95±01.62 | 27.74±01.38 |
| | MAE | 23.63±00.89 | 23.81±00.76 | 25.88±00.32 | 26.10±00.26 | 25.07±00.64 |
| | FCN-ResNet50 | 32.39±01.79 | 34.12±02.01 | 36.14±01.19 | 35.74±01.41 | 36.17±01.41 |
| | SegFormer | 40.55±03.40 | 44.22±03.50 | 42.52±01.97 | 42.34±02.20 | 42.19±02.63 |
| COCO | SAM | 03.13±00.09 | 06.21±00.32 | 05.31±00.09 | 05.85±00.13 | 05.16±00.31 |
| | DINOv2 | **23.24±00.38** | **20.92±00.19** | **28.30±00.77** | **28.99±01.33** | **28.15±01.24** |
| | iBOT | 09.31±00.68 | 09.35±00.47 | 10.91±01.02 | 11.27±00.99 | 11.67±01.09 |
| | CLIP | 11.32±00.31 | 14.47±00.29 | 15.70±00.71 | 13.62±00.78 | 15.13±00.91 |
| | MAE | 8.54±00.46 | 08.60±00.45 | 10.52±00.45 | 10.13±00.42 | 10.71±00.57 |
| | FCN-ResNet50 | 10.08±00.32 | 08.82±00.88 | 12.05±00.33 | 11.71±00.24 | 11.88±00.38 |
| | SegFormer | 18.37±01.13 | 19.69±00.82 | 21.25±00.52 | 19.64±00.49 | 21.16±00.64 |
| PPD | SAM | 83.14±10.36 | 84.82±00.43 | 81.32±17.07 | 84.94±10.11 | 83.75±10.01 |
| | DINOv2 | **92.31±01.84** | **92.85±01.76** | **93.05±02.57** | **89.67±06.43** | **92.61±02.10** |
| | iBOT | 85.06±08.80 | 87.52±06.23 | 86.04±09.37 | 85.48±07.80 | 87.63±08.00 |
| | CLIP | 72.49±10.78 | 78.14±06.81 | 78.28±12.63 | 73.13±11.70 | 73.19±08.69 |
| | MAE | 74.34±10.88 | 77.54±07.95 | 76.90±07.95 | 72.87±12.83 | 73.62±10.59 |
| | FCN-ResNet50 | 70.86±05.91 | 89.31±01.88 | 82.01±08.04 | 76.91±10.29 | 70.84±06.15 |
| | SegFormer | 77.88±07.82 | 82.98±07.00 | 88.99±05.98 | 87.82±03.78 | 90.14±03.72 |

Table 4: Detailed results of mIoU on our three considered datasets in 2-shot semantic segmentation.

| | | Linear | Multilayer | SVF | LoRA | Fine-tuning |
|---|---|---|---|---|---|---|
| Cityscapes | SAM | 38.84±00.38 | 42.71±00.24 | 40.24±01.18 | 40.62±00.29 | 40.63±00.77 |
| | DINOv2 | **56.54±00.75** | **53.36±00.09** | **58.21±01.00** | **58.84±01.75** | **58.60±00.57** |
| | iBOT | 33.25±00.62 | 32.42±00.58 | 35.60±00.85 | 36.37±00.93 | 35.29±00.87 |
| | CLIP | 25.19±01.05 | 28.96±01.62 | 30.85±00.42 | 30.53±01.19 | 19.29±04.90 |
| | MAE | 24.15±00.32 | 24.88±00.51 | 27.68±00.21 | 27.92±00.51 | 26.83±00.32 |
| | FCN-ResNet50 | 36.40±00.57 | 37.89±00.45 | 38.98±00.06 | 38.34±00.28 | 38.97±00.33 |
| | SegFormer | 45.42±00.82 | 50.68±01.12 | 46.88±01.35 | 47.35±01.34 | 46.96±01.72 |
| COCO | SAM | 04.45±00.03 | 09.30±00.43 | 07.92±00.22 | 08.82±00.31 | 08.99±00.47 |
| | DINOv2 | **33.08±00.17** | **30.01±00.57** | **39.91±00.90** | **39.01±01.36** | **40.09±00.68** |
| | iBOT | 13.13±00.81 | 13.13±00.63 | 15.65±00.75 | 16.84±00.78 | 16.84±00.88 |
| | CLIP | 16.03±00.47 | 18.85±00.67 | 22.33±00.46 | 20.40±00.99 | 21.85±00.84 |
| | MAE | 12.20±00.69 | 12.26±00.66 | 15.30±00.27 | 15.26±00.34 | 15.66±00.21 |
| | FCN-ResNet50 | 12.54±00.50 | 11.44±00.48 | 13.23±00.78 | 13.42±00.81 | 12.55±00.83 |
| | SegFormer | 24.23±00.20 | 25.38±00.25 | 26.34±01.34 | 25.62±00.75 | 28.08±00.72 |
| PPD | SAM | 83.62±10.89 | 88.32±04.81 | 91.51±05.99 | 87.89±09.03 | 87.56±07.23 |
| | DINOv2 | **93.96±01.44** | **94.24±02.03** | **94.17±01.68** | **93.33±02.18** | **94.35±00.86** |
| | iBOT | 90.43±05.44 | 93.05±02.63 | 90.79±06.09 | 90.23±06.42 | 91.59±04.22 |
| | CLIP | 76.01±10.95 | 82.58±03.02 | 83.59±05.94 | 77.21±09.11 | 65.92±15.61 |
| | MAE | 74.36±12.50 | 79.05±07.61 | 78.12±11.30 | 77.93±12.92 | 74.97±10.59 |
| | FCN-ResNet50 | 74.01±05.93 | 92.74±02.02 | 93.76±02.01 | 87.88±06.22 | **94.91±00.59** |
| | SegFormer | 82.74±04.82 | 90.67±06.62 | 90.64±04.80 | 85.66±02.75 | 91.39±01.59 |

Table 5: Detailed results of mIoU on our three considered datasets in 5-shot semantic segmentation.

| | | Linear | Multilayer | SVF | LoRA | Fine-tuning |
|---|---|---|---|---|---|---|
| Cityscapes | SAM | 41.15±01.71 | 44.90±00.74 | 42.29±00.32 | 44.58±00.68 | 45.15±00.68 |
| | DINOv2 | **59.71±02.30** | **59.22±02.16** | **61.22±01.97** | **61.69±01.37** | **62.88±01.74** |
| | iBOT | 36.45±00.24 | 36.67±00.22 | 39.48±00.24 | 41.05±00.72 | 40.46±00.47 |
| | CLIP | 28.70±01.20 | 31.21±01.62 | 33.88±01.23 | 33.23±00.78 | 13.40±01.31 |
| | MAE | 27.49±00.72 | 28.19±01.01 | 30.67±00.91 | 30.92±00.96 | 29.91±01.03 |
| | FCN-ResNet50 | 37.68±00.97 | 39.85±00.13 | 42.42±00.17 | 42.80±00.43 | 43.30±00.75 |
| | SegFormer | 48.26±01.13 | 53.39±01.32 | 48.00±00.75 | 50.06±00.49 | 50.00±00.63 |
| COCO | SAM | 05.92±00.17 | 14.01±00.05 | 11.89±00.72 | 14.30±00.45 | 14.92±00.76 |
| | DINOv2 | **44.48±00.64** | **42.32±00.66** | **49.07±01.59** | **45.82±01.80** | **50.52±01.34** |
| | iBOT | 20.99±00.81 | 20.85±00.61 | 24.33±00.96 | 27.62±00.88 | 27.28±01.20 |
| | CLIP | 26.71±00.57 | 29.11±00.67 | 31.85±00.15 | 30.71±00.11 | 31.61±00.39 |
| | MAE | 19.20±00.52 | 19.27±00.44 | 21.67±00.46 | 22.80±00.40 | 23.39±00.39 |
| | FCN-ResNet50 | 15.42±00.27 | 14.23±00.65 | 15.47±00.23 | 15.94±00.16 | 14.09±00.10 |
| | SegFormer | 31.80±01.20 | 33.47±01.19 | 34.68±01.47 | 32.87±01.23 | 36.56±01.52 |
| PPD | SAM | 92.59±01.81 | **96.28±00.19** | 95.61±00.65 | **95.82±00.58** | 96.65±00.78 |
| | DINOv2 | **95.21±00.35** | 96.13±00.35 | 95.62±00.47 | 95.11±00.39 | 93.06±03.70 |
| | iBOT | 95.02±00.95 | 95.71±00.25 | 95.54±00.63 | 95.34±00.71 | 96.57±00.16 |
| | CLIP | 83.48±01.12 | 85.68±00.43 | 88.94±00.58 | 83.07±01.68 | 85.76±01.16 |
| | MAE | 86.39±00.42 | 87.61±00.31 | 86.26±01.99 | 87.34±00.74 | 86.73±00.36 |
| | FCN-ResNet50 | 87.97±00.79 | 95.02±01.11 | **96.84±00.15** | 95.45±00.65 | **97.13±0.02** |
| | SegFormer | 90.48±00.82 | 95.92±00.67 | 94.56±00.06 | 91.97±00.24 | 94.03±00.19 |

Table 6: Detailed results of mIoU on our three considered datasets in 10-shot semantic segmentation.

| | | Linear | Multilayer | SVF | LoRA | Fine-tuning |
|---|---|---|---|---|---|---|
| Cityscapes | SAM | 42.10±01.24 | 48.41±00.34 | 42.74±00.42 | 46.75±00.27 | 47.17±00.68 |
| | DINOv2 | **63.19±00.23** | **63.64±00.64** | **63.38±00.45** | **63.87±00.46** | **66.24±00.49** |
| | iBOT | 39.47±00.51 | 39.22±00.98 | 40.46±00.80 | 43.05±01.07 | 43.79±01.05 |
| | CLIP | 32.44±01.32 | 34.83±00.82 | 36.76±01.07 | 36.50±01.17 | 21.83±10.21 |
| | MAE | 30.46±00.28 | 30.59±00.18 | 32.63±00.75 | 32.99±00.45 | 31.85±00.55 |
| | FCN-ResNet50 | 40.63±00.66 | 43.20±00.38 | 45.12±00.53 | 45.56±00.62 | 45.63±01.38 |
| | SegFormer | 50.76±00.56 | 56.77±00.66 | 50.12±00.45 | 52.81±00.03 | 52.08±00.17 |
| COCO | SAM | 07.13±00.25 | 17.81±00.33 | 15.02±00.31 | 19.36±00.74 | 21.32±00.49 |
| | DINOv2 | **50.84±00.43** | **49.83±00.34** | **50.84±01.23** | **48.55±00.66** | **52.74±01.09** |
| | iBOT | 28.23±00.46 | 28.30±00.38 | 29.03±00.83 | 35.26±01.08 | 35.20±00.80 |
| | CLIP | 33.75±00.36 | 36.45±00.53 | 35.36±00.54 | 35.38±00.33 | 34.19±00.10 |
| | MAE | 25.38±00.22 | 25.88±00.29 | 20.96±00.54 | 08.84±11.26 | 26.86±00.25 |
| | FCN-ResNet50 | 17.04±00.43 | 16.74±00.46 | 16.81±00.24 | 17.15±00.21 | 15.55±00.17 |
| | SegFormer | 36.47±00.91 | 38.97±00.88 | 37.64±01.66 | 35.59±00.56 | 40.61±01.32 |
| PPD | SAM | 94.15±00.66 | **96.57±00.07** | 96.48±00.14 | **96.49±00.13** | 97.12±00.54 |
| | DINOv2 | **95.55±00.15** | 96.38±00.24 | 95.67±00.53 | 95.32±00.41 | 94.24±02.64 |
| | iBOT | 95.53±00.08 | 96.02±00.13 | 96.01±00.11 | 95.68±00.22 | 96.78±00.11 |
| | CLIP | 84.24±00.51 | 86.18±00.27 | 88.81±00.76 | 83.37±00.74 | 86.84±00.39 |
| | MAE | 86.84±00.74 | 87.44±00.48 | 86.59±00.46 | 87.00±01.90 | 87.17±00.45 |
| | FCN-ResNet50 | 88.40±01.72 | 96.01±00.28 | **97.18±00.20** | 96.34±00.16 | **97.39±00.11** |
| | SegFormer | 91.35±00.13 | 96.49±00.06 | 95.21±00.15 | 92.21±00.30 | 94.68±00.27 |

Table 7: Effect of PEFT methods on mIoU, bold numbers denote top performance across methods.

|            | SVF | LoRA | VPT | Adaptformer | Convpass | Bitfit |
|------------|-----|------|-----|-------------|----------|--------|
| Cityscapes | 51.96±01.37 | **54.35±01.55** | 51.92±00.56 | 50.44±00.53 | 50.24±01.12 | 51.55±00.58 |
| COCO       | 28.30±00.77 | 28.99±01.33 | **30.15±02.10** | 28.68±01.23 | 29.35±01.63 | 29.19±00.94 |
| PPD        | **93.05±02.57** | 89.67±06.43 | 91.25±02.11 | 92.87±01.47 | 92.21±02.50 | 92.91±01.50 |

## A.2 Parameter Efficient Fine-Tuning (PEFT) Methods.

Given the strong performance of Parameter-Efficient Fine-Tuning (PEFT) methods on our benchmark, we opted to evaluate various PEFT approaches from multiple families. We examined LoRA and SVF alongside two adapter tuning methods: AdaptFormer (Chen et al., 2022), which integrates a lightweight adapter module parallel to the feed-forward network within the encoder blocks and only trains these adapters, and Convpass (Jie & Deng, 2022), which adds convolutional layers to the parallel adapter to incorporate inductive biases and trains only these layers. Additionally, we assessed Visual Prompt Tuning (VPT) (Jia et al., 2022), which introduces tokens at the input of encoder blocks and trains only these tokens, and a partial-based tuning method, BitFit(Ben Zaken et al., 2022), which freezes the model weights and fine-tunes only the biases. For computing constraints, we only tested these methods with DINOv2 on 1-shot setting.

The results, presented in Table 7, indicate minimal differences among the PEFT methods, suggesting no definitive superior method. This outcome underscores the critical role of feature extractors in determining the effectiveness of these tuning approaches.

## A.3 SAM Underperformance

In this section, we delve into the reasons behind SAM's poor performance on the COCO dataset, exploring two main hypotheses:

- **Decoder Limitations:** The primary suspect for SAM's subpar performance is the dependency of the image encoder on the prompt encoder and the mask decoder. To address this, we replaced the original mask decoder with a transformer decoder, as detailed in (Kirillov et al., 2023), modifying it for few-shot semantic segmentation following the inspiration from (Zhong et al., 2024). We introduced a constant prompt to the decoder, altering tokens to produce an $N$-class depth map rather than binary masks, and eliminated the IoU score output branch. We experimented with two mask decoder configurations, one with a Conv Transpose module and another where we upscaled the output through interpolation, while initializing remaining modules with pre-trained weights. Despite these adjustments, this approach was outperformed by a simpler linear decoding method by up to 5% mIoU, likely due to the challenges of training a complex decoder in a few-shot scenario.

- **Mask Distribution Bias:** Another investigated issue was the mask distribution bias in COCO towards image centers, unlike the distribution in the SA-1B dataset used for SAM, highlighted in (Kirillov et al., 2023), where masks are more evenly distributed. We attempted to mitigate this bias with various preprocessing techniques (RandomCrop, CenterCrop, dividing each image into four smaller images); however, these modifications did not enhance our results.

These findings shed light on the specific challenges faced by SAM in adapting to the COCO dataset and highlight the complexities involved in optimizing for few-shot semantic segmentation tasks.

## A.4 Practical Considerations

Previous values were obtained by selecting the best learning rates on the validation sets. Instead, here we study the effect of transfering the learning rate from a dataset to another one on the mIoU. The learning rate values we explored are $10^{-2}$, $10^{-3}$, $10^{-4}$, $10^{-5}$ and $10^{-6}$. Table 8 shows the drop of mIoU when we train a pair (model, method) on dataset A with the best learning rate for dataset B. We can see that the learning rate is relatively consistent between the 3 datasets and especially between Cityscapes and COCO for a given model and a given method. Even though the optimal values are not exactly the

Table 8: Drop in mIoU when using the best learning rate corresponding to the dataset on the line (source) when training on the dataset on the column (target), instead of using the target validation set to find the best learning rate.

| | | SVF | | | LoRA | | | Fine-tuning | | |
|---|---|---|---|---|---|---|---|---|---|---|
| | | Cityscapes | COCO | PPD | Cityscapes | COCO | PPD | Cityscapes | COCO | PPD |
| SAM | Cityscapes | - | 0 | 0 | - | 0 | 0 | - | 02.72 | 22.83 |
| | COCO | 0 | - | 0 | 0 | - | 0 | 00.94 | - | 03.01 |
| | PPD | 0 | 0 | - | 0 | 0 | - | 03.38 | 03.83 | - |
| DINOv2 | Cityscapes | - | 0 | 0 | - | 0 | 02.31 | - | 0 | 15.62 |
| | COCO | 0 | - | 0 | 0 | - | 02.00 | 0 | - | 26.30 |
| | PPD | 0 | 0 | - | 00.88 | 00.88 | - | 01.17 | 01.17 | - |
| CLIP | Cityscapes | - | 0 | 00.40 | - | 00.52 | 00.52 | - | 0 | 0 |
| | COCO | 0 | - | 01.09 | 00.24 | - | 0 | 0 | - | 0 |
| | PPD | 01.91 | 01.91 | - | 01.53 | 0 | - | 0 | 0 | - |
| MAE | Cityscapes | - | 00.19 | 00.19 | - | 00.54 | 00.54 | - | 00.12 | 0 |
| | COCO | 00.28 | - | 0 | 00.87 | - | 0 | 00.65 | - | 00.65 |
| | PPD | 01.41 | 0 | - | 01.47 | 0 | - | 0 | 01.53 | - |
| ResNet | Cityscapes | - | 01.07 | 00.23 | - | 00.82 | 0 | - | 00.56 | 0 |
| | COCO | 00.88 | - | 04.07 | 00.66 | - | 00.66 | 01.55 | - | 01.55 |
| | PPD | 08.46 | 18.01 | - | 0 | 07.85 | - | 0 | 12.13 | - |
| iBOT | Cityscapes | - | 0 | 00.37 | - | 0 | 00.52 | - | 09.85 | 09.85 |
| | COCO | 0 | - | 00.35 | 0 | - | 00.84 | 00.65 | - | 0 |
| | PPD | 01.97 | 01.97 | - | 01.67 | 01.67 | - | 02.01 | 0 | - |
| SegFormer | Cityscapes | - | 0 | 0 | - | 0 | 0 | - | 0 | 0 |
| | COCO | 0 | - | 0 | 0 | - | 0 | 0 | - | 0 |
| | PPD | 0 | 0 | - | 0 | 0 | - | 0 | 0 | - |

same between these datasets, the variation of performance is low and applying the optimal learning of one dataset on the other does not cause a big drop of performance, so we argue that these values could serve as a good baseline for a new unseen dataset. The higher drops occur when we either take the optimal learning rate for PPD or when we test an optimal learning rate of Cityscapes or COCO on PPD. This is mainly due to the fact that the support set of PPD is significantly smaller than that of the other considered datasets.

## A.5 More Individual Factors

### A.5.1 Input Resolution Effect.

We explore how input resolution impacts model performance, a critical factor given our training setup. Specifically, CLIP and MAE were trained at a resolution of 224x224, while other models were trained with a higher resolution of 1024x1024. This discrepancy is due to the fixed input size requirements of CLIP and MAE's positional embeddings. To address this, we adapted the positional embeddings matrix for variable resolutions, a common practice for ViT models. Our findings, illustrated in Figure 3, reveal a divergence in performance trends: increasing resolution degrades results for CLIP and MAE but improves them for DINOv2. This can be attributed to DINOv2's training across multiple resolutions, unlike CLIP and MAE. Consequently, this suggests using 1024x1024 input images on DINOv2 whereas 224x224 input images on CLIP and MAE is fair for performance comparison.

### A.5.2 Registers Effect.

In the study by (Darcet et al., 2023), it is demonstrated that DINOv2, unlike its predecessor DINOv1, presents artifacts within its feature maps. This phenomenon is attributed to the presence of tokens with unusually high norms, which are integral to the model's internal computations. To address this issue, the authors introduced a modified version of DINOv2, incorporating additional tokens termed "Registers" during the training phase, which are subsequently omitted during inference. As evidenced in Table 9, this

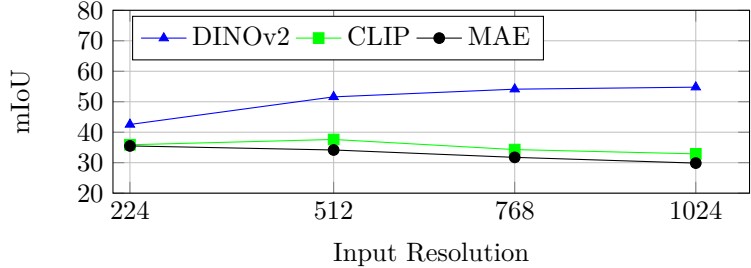

Figure 3: Impact of the input resolution on the obtained mIoU.

Table 9: Effect of training with registers on mIoU

|  | Linear | | | LoRA | | |
|---|---|---|---|---|---|---|
|  | Cityscapes | COCO | PPD | Cityscapes | COCO | PPD |
| DINOv2 w/o Registers | **48.80±01.82** | 23.24±00.38 | **92.31±01.84** | **54.35±01.55** | 28.99±01.33 | 89.67±06.43 |
| DINOv2 w/ Registers | 48.06±02.46 | **27.42±00.72** | 91.58±02.41 | 54.08±00.44 | **34.02±02.26** | **91.03±02.03** |

innovative approach significantly enhances DINOv2's performance on the COCO dataset, showcasing the potential benefits of Registers during training for FSS tasks.

## A.6 Model Comparison Summary

Table 10 summarizes the main components of the different models used in our study, we discuss the importance of each component (Architecture, Size, Pre-training Dataset, Pre-training Method) in 6.3.

## A.7 Dataset Comparison Summary

Table 11 and Figure 4 provide an overview of the three datasets considered in this study. These datasets exhibit varying levels of class diversity. Notably, in the PPD dataset, segmentation masks show significant variation across images, contributing to a higher standard deviation in mIoU. This variability poses a challenge in one-shot learning scenarios, where models must generalize effectively from only two reference images.

## A.8 Impact on the Number of Shots

We can see in Fig. 5 that for SVF, the different models scale similarly and that models are still outperformed by DINOv2 even when more shots are available. We can see in Fig. 6 that for DINOv2, the gap between the models tends to be smaller the more we increase the number of shots. Also, full fine-tuning scales better than other methods because we are less prone to overfitting, leading to the conclusion that the is it better to choose the full fine-tuning method when more shots are available. Yet it is not crucial since the full fine tuning is time and memory expensive compared to the other methods, and a simple linear probing can give good performance with a minor drop in performance.

Table 10: Summary of the main model elements.

|  | Architecture | Size | Dataset | Pre-training Method |
|---|---|---|---|---|
| SAM | ViT | 89M | SA-1B | MAE + Prompt Segmentation |
| DINOv2 | ViT | 86M | 142M image dataset | DINO (SSL with Knowledge Distillation) |
| iBOT | ViT | 85M | ImageNet-1K | Masked image modeling + knowledge distillation |
| CLIP | ViT | 86M | 400M of (image,text) pairs | image, text embedding alignement |
| MAE | ViT | 86M | ImageNet-1K | Masking + Image reconstruction |
| FCN-ResNet50 | CNN | 23M | Subset of COCO | Semantic Segmentation |
| SegFormer | ViT | 81M | ADE20K | Semantic Segmentation |

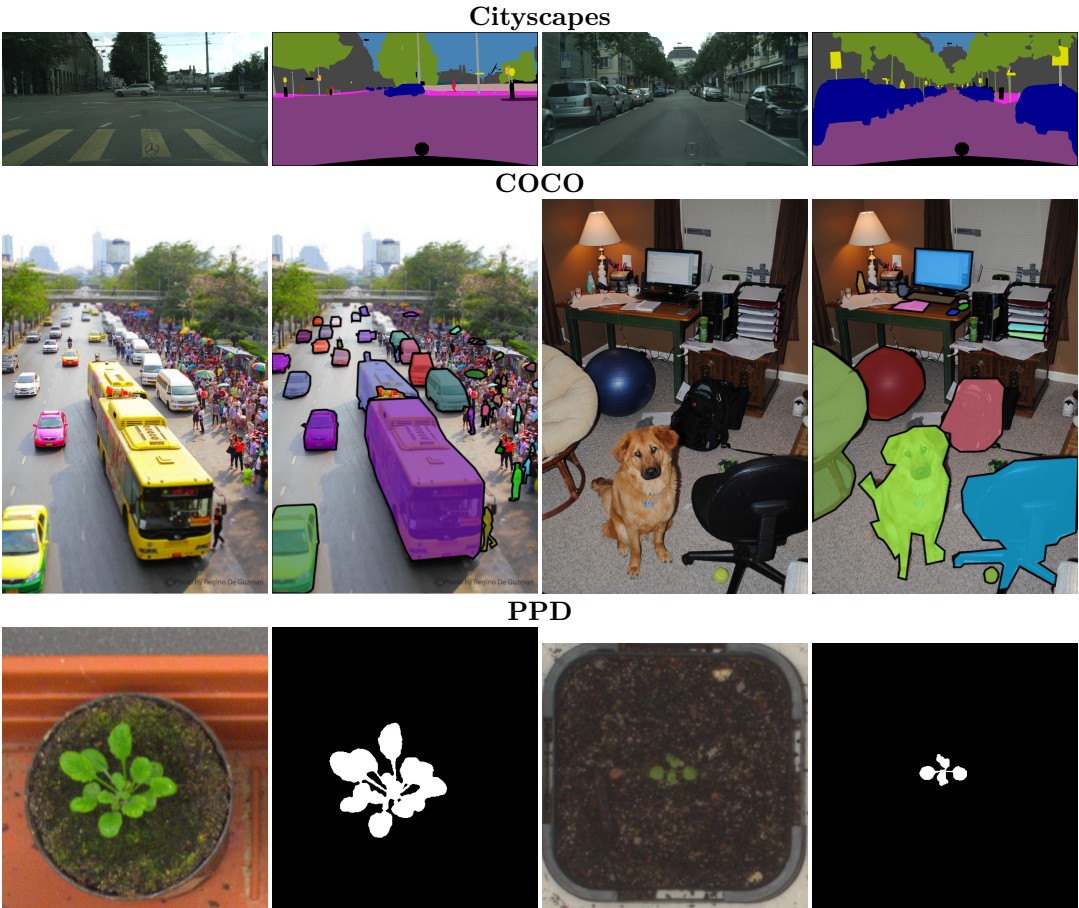

Figure 4: Examples of images and corresponding segmentation masks from Cityscapes, COCO, and PPD datasets.

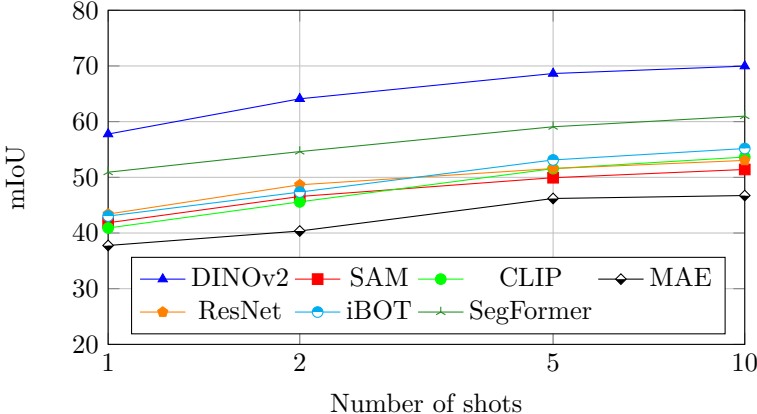

Figure 5: Impact of the number of shots on mIoU for the SVF method.

Table 11: Overview of the datasets used in our benchmark.

| Dataset | Number of Classes | Resolution | Size of Training Set |
|---|---|---|---|
| Cityscapes | 19 | 1024×2048 | 2,975 |
| COCO | 80 | (400–640)x(400–640) | 118k |
| PPD | 2 | 500×500 | 783 |

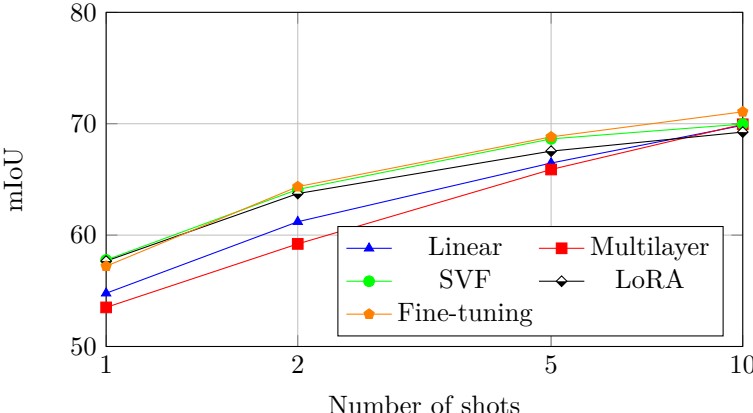

Figure 6: Impact of the number of shots on mIoU for DINOv2.

## A.9 COVID-19 CT Dataset

We conducted additional experiments on a fourth dataset, a medical dataset for COVID-19 CT image segmentation (https://www.kaggle.com/c/covid-segmentation), which includes 4 classes. The 1-shot results (mIoU), presented below, demonstrate a significant performance drop for Vision Foundation Models (VFMs) trained predominantly with SSL methods on natural images when applied to medical images. This gap in performance, caused by the substantial distributional differences between natural and medical images, highlights the advantages of current semantic segmentation models, such as ResNet-50 and SegFormer, in handling tasks with large domain shifts.

Table 12: mIoU averaged over our three considered datasets in 1-shot semantic segmentation for COVID-19 CT Dataset. Bold numbers correspond to the best performance across extractors and underline numbers to the best performance across adaptation methods.

| Model | Linear | Multilayer | SVF | LoRA | Finetune |
|---|---|---|---|---|---|
| SAM | 23.20±02.37 | 45.69±05.40 | 32.90±03.21 | 39.28±05.25 | 24.49±01.39 |
| DINOv2 | 34.74±02.52 | 40.18±02.35 | 30.11±07.36 | 34.05±00.76 | 32.01±04.82 |
| CLIP | 34.95±01.45 | 35.50±01.17 | 36.70±02.40 | 34.71±00.83 | 26.98±01.61 |
| MAE | 23.37±01.53 | 24.10±01.75 | 23.48±01.83 | 23.92±01.75 | 21.04±01.22 |
| FCN-ResNet50 | **44.01±01.27** | **54.03±01.14** | **56.97±01.05** | **54.71±01.41** | **57.35±02.14** |
| IBOT | 23.51±03.19 | 24.54±03.02 | 21.87±00.46 | 40.28±03.79 | 21.49±01.63 |
| SEGFORMER | 44.14±00.82 | 50.30±01.20 | 49.35±01.02 | 47.51±01.49 | 52.25±00.99 |

## A.10 Limitations

While our benchmark introduces novel insights into the adaptation of Vision Foundation Models (VFMs) for few-shot semantic segmentation, several limitations warrant further investigation:

- **Dataset Representation:** Our benchmark relies on a limited number of datasets, primarily from urban scenes (Cityscapes), natural images (COCO), and plant phenotyping (PPD). Although diverse, this selection does not encompass other important domains, such as satellite imagery or underwater scenes, where few-shot segmentation could be valuable.

- **Model Adaptation Scope:** The study focuses on a specific set of pretrained VFMs and adaptation methods. While comprehensive, this excludes potential alternative models, such as specialized medical imaging networks or models pretrained on multimodal datasets, which could perform better in domain-specific settings.

- **Domain Gap:** The significant drop in performance observed for medical images (COVID-19 CT dataset) highlights that VFMs trained on natural images are not inherently robust to large domain shifts. Addressing this limitation may require dedicated pretraining or domain adaptation strategies, which were outside the scope of this study.

- **Energy Efficiency:** The absence of an energy efficiency analysis is a limitation of our work. However, it is evident that methods like LoRA and SVF are considerably more efficient than full fine-tuning, as they require fewer computational resources and less processing time while maintaining strong performance. A more detailed study on the energy consumption of different adaptation techniques would be valuable for future research.

These limitations present opportunities for future research to enhance the applicability and robustness of VFMs in few-shot semantic segmentation, particularly in challenging, real-world scenarios.

### A.11 Negative Societal Impact

VFMs are trained on carefully curated large-scale datasets designed to exclude harmful content and private information, mitigating privacy risks a priori. These datasets are generally less biased than traditional datasets, helping reduce dataset-induced biases in model predictions. However, potential biases and harmful content may still arise, necessitating continuous efforts in debiasing and content filtering. Additionally, while the benchmark we developed has minimal potential for direct malicious applications, the models used in our research could still reflect biases from their pretraining or adaptation datasets. Furthermore, by streamlining semantic segmentation and enabling effective model training with limited labeled data, this approach could inadvertently lower the barrier for applications in surveillance or other privacy-sensitive domains. While this accessibility fosters innovation and practical deployment, it also highlights the importance of responsible use and potential regulatory considerations in sensitive contexts.

### A.12 Object Size Effect on mIOU

We analyzed the relationship between object size and segmentation performance by plotting the mIoU for the Car class against its pixel area (see Figure 7). Our results show that larger cars tend to achieve higher mIoU scores, while smaller cars exhibit lower mIoU and higher variance.

This trend may not solely reflect better model performance but could be due to a geometric bias: larger objects have more inner pixels, which are easier to segment, whereas smaller objects have a higher proportion of boundary pixels, which are harder to segment accurately.

These findings suggest that object size influences segmentation performance, but the apparent advantage for larger objects may stem from intrinsic biases rather than true model generalization.

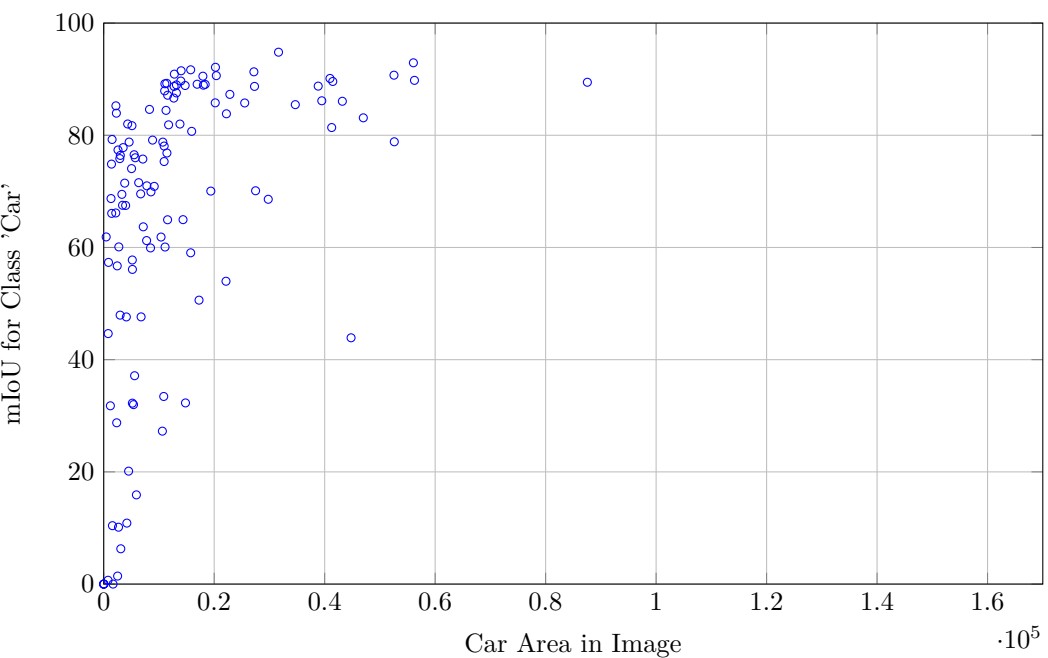

Figure 7: Impact of Car Area on mIoU for the 'Car' class using DINOv2.

