# OpenReview forum: "A Novel Benchmark for Few-Shot Semantic Segmentation in the Era of Foundation Models"
_TMLR — Accepted by TMLR_

### Review · Reviewer_gkw8 · 2025-02-13

**Summary Of Contributions:**

This paper proposes a new benchmark for FSS to evaluate the performance of VFM in FSS tasks. The authors conducted detailed experimental analysis around multiple VFMs (SAM, MAE, DINOv2, CLIP) using this benchmark and tried various adaptation methods, including linear probing, PEFT, full fine-tuning. The authors found that DINOv2 performed the best on this benchmark, even outperforming segmentation-based models.

**Audience:**

Yes

**Claims And Evidence:**

Yes

**Requested Changes:**

Please see Weaknesses section for the requested changes.

**Strengths And Weaknesses:**

Strengths
1. The paper is well-written and easy to follow.
2. The paper proposes a new benchmark and introduces the process of constructing the benchmark. Unlike the previous FSS setting that only supports single-class binary masks, this benchmark supports multi-class masks.
3. The experiments are very comprehensive and detailed. The authors conducted experiments on multiple VFMs, including SAM, MAE, DINOv2, CLIP, and tried various adaptation methods, including linear probing, PEFT, full fine-tuning. They also explored the impact of other factors such as model size.

Weaknesses
1. Regarding the task setting, it seems that Generalized Few-shot Semantic Segmentation[1] (GFSS) can also support multi-class, and it needs to make a clearer comparison between the setting of this paper and traditional FSS, GFSS in the task formulation.
[1] Generalized few-shot semantic segmentation. CVPR 2022

2. Recently, there have been some works using VFM for FSS, and it would be better to have a more detailed discussion of these works in the related work.
For example, some works use a single VFM such as CLIP, SAM, DINOv2 for FSS, some works combine multiple VFMs for FSS, and some works explore the use of generative VFMs(SD) for FSS.

[2] Visual and Textual Prior Guided Mask Assemble for Few-Shot Segmentation and Beyond (based on CLIP)

[3] Matcher: Segment Anything with One Shot Using All-Purpose Feature Matching
(DINOv2+SAM) ICLR 2024

[4] VRP-SAM: SAM with visual reference prompt. (based on SAM) CVPR2024

[5] Segic: Unleashing the emergent correspondence for in-context segmentation. (based on DINOV2) ECCV 2024

[6] Unleashing the potential of the diffusion model in few-shot semantic segmentation. (based on SD) Neurips 2024

[7] Bridge the Points: Graph-based Few-shot Segment Anything Semantically. (DINOv2+SAM) Neurips 2024

3. The explanation of the adaptation methods is not clear enough. For example, about LORA, "Our procedure begins with training the decoder using the linear method and then introduces LoRA adapters to the encoder. Subsequently, we fine-tune both the adapters and decoder parameters." Is this means that there are three stages, the first stage is linear probing, the second stage is LoRA adapters to the encoder, and the third stage is fine-tune both the adapters and decoder parameters. Is it reasonable to divide it into three stages? Is it feasible to only perform the third stage?

---

> ### Author Response · Authors · 2025-03-04
> **Response to reviewer gkw8**
>
> We thank the reviewer for taking the time to read our paper and for the helpful comments.
>
> >Regarding the task setting, it seems that Generalized Few-shot Semantic Segmentation[1] (GFSS) can also support multi-class, and it needs to make a clearer comparison between the setting of this paper and traditional FSS, GFSS in the task formulation
>
> Indeed, GFSS supports multi-class segmentation and shares similarities with our task formulation. However, a key distinction lies in the assumption of base classes in GFSS, which differs from foundation models trained predominantly using self-supervised learning (SSL) methods. Unlike GFSS, our benchmark does not rely on base classes, making it incompatible with traditional GFSS settings. The introduction has been revised to clarify this distinction and explicitly account for GFSS.
>
> >Recently, there have been some works using VFM for FSS, and it would be better to have a more detailed discussion of these works in the related work. For example, some works use a single VFM such as CLIP, SAM, DINOv2 for FSS, some works combine multiple VFMs for FSS, and some works explore the use of generative VFMs(SD) for FSS.
>
> We thank the reviewer for pointing out these relevant references. The cited recent works have explored the use of vision foundation models (VFMs) such as CLIP, DINOv2, and SAM for FSS, adapting these models to the FSS setting. However, these studies primarily focus on specific model adaptations rather than providing a comparative analysis of their effectiveness as feature extractors. Our objective is to establish a new benchmark, as we argue that existing ones are not well suited for comparing VFMs, and to provide a baseline with multiple straightforward methods. A paragraph has been added in the Related Work section (Section 2.2 on FSS) to discuss these recent works and how they complement our benchmark and comparative study.
>
> >he explanation of the adaptation methods is not clear enough. For example, about LORA, "Our procedure begins with training the decoder using the linear method and then introduces LoRA adapters to the encoder. Subsequently, we fine-tune both the adapters and decoder parameters." Is this means that there are three stages, the first stage is linear probing, the second stage is LoRA adapters to the encoder, and the third stage is fine-tune both the adapters and decoder parameters. Is it reasonable to divide it into three stages? Is it feasible to only perform the third stage?
>
> For LoRA, SVF, and full fine-tuning, the training procedure consists of two stages. In the first stage, we train the classifier (decoder) using the linear method. In the second stage, we train both the decoder and adapters (or all weights in the case of full fine-tuning). The motivation for using two stages is the significant difference in gradient magnitudes between the decoder and the encoder[1]. Since the classifier is trained from scratch while the encoder is only fine-tuned or adapted, a staged approach helps stabilize training. An alternative would be a single-stage approach with different learning rates for the encoder and decoder, but we opted for two stages as the first stage aligns with the linear probing step already included in our methodology. We rewrote parts of the “training procedure” part to clarify this point. (Subsection 6.1)
>
> [1]Fine-Tuning can Distort Pretrained Features and Underperform Out-of-Distribution

---

### Review · Reviewer_dmLk · 2025-02-14

**Summary Of Contributions:**

This paper introduces a new benchmark for evaluating Few-Shot Semantic Segmentation (FSS) performance in the context of Vision Foundation Models (VFMs). The benchmark is designed to overcome limitations in prior FSS benchmarks by:

- Incorporating fully labeled few-shot examples without assuming class balance.
- Evaluating five foundation models (DINOv2, SAM, CLIP, MAE, iBOT) with five adaptation strategies (linear probing, multilayer feature extraction, singular value fine-tuning (SVF), LoRA, and full fine-tuning).
- Analyzing how different pretraining strategies (supervised vs. self-supervised) impact FSS performance.
- Demonstrating that self-supervised models (DINOv2) outperform models explicitly designed for segmentation.

The findings suggest that feature extractor quality is more important than segmentation-specific design and that parameter-efficient fine-tuning methods like LoRA and SVF can match the performance of full fine-tuning.

**Audience:**

Yes

**Broader Impact Concerns:**

- If VFMs have been trained on biased datasets, their performance in few-shot segmentation may carry dataset-induced biases. A discussion on fairness implications should be included.

- If foundation models memorize specific training data, they could pose privacy risks when used in FSS. A short discussion on data leakage risks should be added.

- Many VFMs (e.g., CLIP, DINOv2) are trained on web-scale datasets without explicit human oversight. How do these models handle harmful content or biases in their few-shot predictions?

- Full fine-tuning is computationally expensive compared to LoRA or SVF. The paper should briefly discuss energy efficiency as a practical consideration.

**Claims And Evidence:**

Yes

**Requested Changes:**

I believe that major revisions required:

- Clarify the potential data leakage issue: Discuss whether pretraining datasets contain test classes, and how this affects model performance.
- Justify the exclusion of meta-learning approaches: Many few-shot learning benchmarks include methods like ProtoNets and MAML. At least discuss why they were not considered.
- Expand evaluation to large-scale noisy datasets: Would the proposed benchmark still be reliable on noisier datasets like LAION-400M?
- Investigate why PPD results are so high: Provide a comparison of dataset complexity to explain why PPD leads to much higher mIoU scores.

I also suggest the authors provide additional implementation details:
- How were fine-tuning hyperparameters set?
- Did learning rates vary across layers?
- Were batch normalization statistics frozen?
- Expand discussion on adaptation vs. feature extractor quality: The paper finds that DINOv2 outperforms segmentation-specific models. Could this be due to DINOv2’s pretraining dataset rather than its architecture?
Include error bars or confidence intervals in Figure 1 to indicate result stability across runs.

**Strengths And Weaknesses:**

**Strengths**

+ The study is well-motivated, given the rising adoption of VFMs in vision tasks.

+ The paper systematically compares five VFMs and five adaptation techniques across three datasets (COCO, Cityscapes, PPD), providing a thorough empirical analysis.

+ Unlike traditional FSS benchmarks that focus on in-domain generalization, this benchmark emphasizes adaptation to new distributions while addressing issues like class imbalance and multi-class labeling.

+ The finding that self-supervised models (DINOv2) outperform segmentation-focused models (SAM, CLIP) is non-trivial and challenges common assumptions about FSS.

+ Ablation studies: The authors provide thorough ablation studies to investigate the impact of dataset size, adaptation strategy, and feature extractor depth.

**Weaknesses**

- Benchmark novelty is somewhat incremental. While the benchmark improves upon existing ones (e.g., PASCAL-5i, COCO-20i), it inherits many design choices from prior FSS benchmarks, making it an evolution rather than a fundamental shift.

- Missing large-scale noisy dataset evaluation: The benchmark is tested on COCO, Cityscapes, and PPD, but does not evaluate generalization on large-scale noisy datasets (e.g., LAION-400M, WIT-400M), which are common in real-world applications.

- Lack of discussion on pretraining dataset overlap: Some VFMs (e.g., CLIP, DINOv2) are trained on massive web-scale datasets. How does the presence of test classes in these pretraining datasets affect the results? This is not explicitly addressed.

- Lack of discussion on meta-learning approaches: Few-shot learning often leverages meta-learning strategies (e.g., ProtoNets, MAML, MetaOptNet). Why were these omitted from the adaptation methods?

Also I believe that implementation details need more clarity:

- How were learning rates set across layers in full fine-tuning?
- Were batch norm statistics frozen or updated?
- How does LoRA compare in terms of runtime vs. accuracy tradeoff?
- PPD dataset results seem unusually high: DINOv2 achieves ~90% mIoU on PPD, significantly higher than on COCO or Cityscapes. Is this due to dataset bias or overfitting? A discussion is needed.

---

> ### Author Response · Authors · 2025-03-04
> **Response to reviewer dmLk (1/2)**
>
> We thank the reviewer for taking the time to read our paper and for the helpful comments.
>
>
> >Clarify the potential data leakage issue: Discuss whether pretraining datasets contain test classes, and how this affects model performance.
>
> -We noted in the paper that Cityscapes was part of the pretraining dataset for DINOv2 (in subsection 6.2), meaning the model has encountered images from a similar distribution in an SSL setting, though without segmentation masks. For other models like CLIP and SAM, assessing dataset overlap is more challenging since their pretraining datasets are either private or consist of scraped images, making it difficult to determine similarity with our datasets. While we acknowledge that such data leakage may lead to an unfair comparison, our primary objective, as stated in the introduction, is to evaluate the adaptation of off-the-shelf vision foundation models rather than their generalization abilities. Similar considerations apply to existing benchmarks such as CoOp (Learning to Prompt for Vision-Language Models), which focus on adapting CLIP without addressing dataset overlap explicitly. Recent works have started analyzing this issue [1]. We changed the text to better reflect this potential issue.
>
> [1] PRE-TRAINING CONCEPT FREQUENCY IS PREDICTIVE OF CLIP ZERO-SHOT PERFORMANCE
>
> >Justify the exclusion of meta-learning approaches: Many few-shot learning benchmarks include methods like ProtoNets and MAML. At least discuss why they were not considered.
>
> Meta-learning methods such as ProtoNets, MAML, and MetaOptNet rely on the presence of base classes to train a model in a way that facilitates rapid adaptation to new tasks, typically through episodic training. However, our setting does not include base classes; the model is trained directly on the support set and evaluated on the query set. This fundamental difference makes meta-learning approaches incompatible with our setup. A brief justification for the exclusion of these methods has been added in Section 5.2 about methods.
>
> >Expand evaluation to large-scale noisy datasets: Would the proposed benchmark still be reliable on noisier datasets like LAION-400M?
>
> The dataset LAION-400M does not contain segmentation labels, only image captions, making it unsuitable for our benchmark. Additionally, using large-scale datasets is not necessarily beneficial in our few-shot setting, as the training set is only used to sample support images. To account for noise, we included PPD, a small but highly noisy dataset, where this noise is reflected in the standard deviation of the results. The benchmark includes three datasets with varying scales, resolutions, and numbers of classes to ensure diversity and enable a fair comparison between different models and methods.
>
> >Investigate why PPD results are so high: Provide a comparison of dataset complexity to explain why PPD leads to much higher mIoU scores.
>
> PPD results are high because the benchmark focuses on a relatively simple task—binary plant segmentation—where only two classes (foreground and background) are present. To provide a clearer comparison of dataset complexity, we added a subsection in the appendix (A.7) detailing the characteristics of each dataset, along with a figure illustrating sample images and masks from different datasets. Note that depending on the application, a mIoU of 90% is not necessarily “good”.
>
> -We added more details in the experiment section (Section 6.1 Training Procedure) for the next points:
>
> >How were fine-tuning hyperparameters set?
>
> We performed a grid search to determine the optimal learning rate for PEFT and fine-tuning methods. For linear and multilayer methods, we used a standard learning rate commonly applied to these datasets. Other hyperparameters were set based on values used in previous works, as detailed in the training procedure (Section 6.1).
>
> >Did learning rates vary across layers?
>
> The learning rate remains the same across all layers. Since the classifier is newly initialized, we first train it using the linear method with the best learning rate found for linear probing. Then, we fine-tune both the classifier and the model together using a different learning rate to adjust both the decoder and classifier.
>
> >Were batch normalization statistics frozen?
>
> Most of the models used are ViTs, which do not have batch normalization layers. For ResNet-based models, batch normalization statistics are updated using the support set and frozen for the test phase (query set).

---

> > ### Author Response · Authors · 2025-03-04
> > **Response to reviewer dmLk (2/2)**
> >
> > >Expand discussion on adaptation vs. feature extractor quality: The paper finds that DINOv2 outperforms segmentation-specific models. Could this be due to DINOv2’s pretraining dataset rather than its architecture? Include error bars or confidence intervals in Figure 1 to indicate result stability across runs.
> >
> > In Section 6.3 (Individual Factors), we discuss various elements that could contribute to the strong performance, such as the pretraining dataset and architecture. While a full ablation study on DINOv2 is not feasible, we leverage different available versions of these models as a proxy to assess the impact of various components.
> >
> > >If VFMs have been trained on biased datasets, their performance in few-shot segmentation may carry dataset-induced biases. A discussion on fairness implications should be included.
> >
> > >If foundation models memorize specific training data, they could pose privacy risks when used in FSS. A short discussion on data leakage risks should be added.
> >
> > >Many VFMs (e.g., CLIP, DINOv2) are trained on web-scale datasets without explicit human oversight. How do these models handle harmful content or biases in their few-shot predictions?
> >
> > Foundation models (VFMs) are trained on carefully curated large-scale datasets that are designed to exclude harmful content and private information, thus mitigating privacy risks a priori. These curated datasets are generally less biased than traditional datasets, which helps reduce dataset-induced biases in the model’s predictions. While these models are trained on diverse data, potential biases and harmful content may still arise. However, continuous efforts in debiasing techniques and content filtering are employed to address these challenges. We modified appendix A.11 on negative societal impact accordingly.
> >
> > >Full fine-tuning is computationally expensive compared to LoRA or SVF. The paper should briefly discuss energy efficiency as a practical consideration.
> >
> > We acknowledge that full fine-tuning is computationally expensive compared to more efficient approaches like LoRA or SVF. In practice, these alternative methods offer significant energy efficiency benefits, as they require fewer resources and less computational time while still achieving competitive performance. As energy consumption becomes an increasingly important consideration in machine learning, methods like LoRA and SVF provide a more sustainable solution for model adaptation without compromising on the quality of the results. We added a paragraph in appendix A.10 about limitations

---

### Review · Reviewer_B7hM · 2025-02-18

**Summary Of Contributions:**

The submission contributions are summarized as follows.

**Sufficient Experimental Explorations**. The submission provides sufficient empirical results, exploring potentials of multiple foundation models on few-shot segmentation, including CLIP, SAM, DINO, MAE, iBOT. Such explorations could provide a good reference to the future few-shot segmentation studies.

**New Experimental Setups**. Existing few-shot segmentation approaches mainly benchmark on Pascal-5i or COCO-20i. While, this paper crafts few-shot setups on Cityscapes and PPD, which may provide new testbed for FSS that are more challenging.

**Audience:**

Yes

**Broader Impact Concerns:**

No ethical concern for this paper.

**Claims And Evidence:**

Yes

**Requested Changes:**

### Motivation
**(1-A)** The paper made a claim as follows: "The few-shot examples in our approach are fully labeled, i.e., masks are not necessarily binary. This aspect aligns more closely with contemporary industrial requirements, and can be used with any semantic segmentation
dataset." Authors are suggested to state more clearly on why binary masks are not suited to industrial scenarios, given the fact that most influential papers in this domain apply binary masks.
**(1-B)** The submission made a claim as follows: "Class balance is not assumed, variations are permitted following the distribution of classes within the selected datasets, better reflecting real-world scenarios." The class imbalance is undoubtedly not expected in training distribution, which is well-known as long-tail problem. However, for a benchmark, balance is expected to perform a comprehensive evaluation. While this is more natural, authors seem considering "long-tail" as a good thing for benchmarking, which is quite confusing.

### Experiments
**(2-A)** Given that Stable Diffusion (SD) is also an important vision foundation models and a few existing studies have explored on using SD to perform few-shot segmentation [1-2], authors are suggested to include related experiments and discussions in their submission.
**(2-B)** The proposed setup has its own significance. One aspect is that data like Cityscapes are diverse in object size. For example, a close-up of car compared to a car far away from camera, where instances of car vary in resolution (a.k.a, size). This might be a good perspective to make this paper more solid. Authors are suggested to include such case studies in this paper.
**(2-C)** Notice that authors use resolution of 224 for CLIP and MAE, which may make the few-shot segmentation on Cityscapes quite challenging, as the typical input resolution for such data is 1024. This may lead to false conclusion for design choices when performing few-shot segmentation. Authors are suggested to use OpenCLIP ConvNeXT-L, with input resolution of 1120px as presented in this paper [3].

### Citations

[1] DifFSS: Diffusion Model for Few-Shot Semantic Segmentation.
[2] Unleashing the Potential of the Diffusion Model in Few-shot Semantic Segmentation. NeurIPS 2024.
[3] Convolutions Die Hard: Open-Vocabulary Segmentation with Single Frozen Convolutional CLIP. NeurIPS 2023.

**Strengths And Weaknesses:**

**Strength**. The experiments are sufficient and provide empirical evidences for future few-shot segmentation works, which is a good point.

**Weakness**. The paper needs to be more accurate on claiming contributions. Motivations are not strong enough. A few experiments should be involved as well.

For weaknesses, please refer to **Requested Changes** part.

---

> ### Author Response · Authors · 2025-03-04
> **Response to reviewer B7hM (1/2)**
>
> We thank the reviewer for taking the time to read our paper and for the helpful comments.
>
> ### Motivation :
>
> > (1-A) The paper made a claim as follows: "The few-shot examples in our approach are fully labeled, i.e., masks are not necessarily binary. This aspect aligns more closely with contemporary industrial requirements, and can be used with any semantic segmentation dataset." Authors are suggested to state more clearly on why binary masks are not suited to industrial scenarios, given the fact that most influential papers in this domain apply binary masks.
>
> Most influential papers in semantic segmentation use fully labeled masks. However, in the context of FSS, current benchmarks typically rely on binary masks, as they are inspired by the N-way K-shot setting from few-shot classification. In segmentation, if masks are fully labeled, defining a strict N-way K-shot setting becomes challenging because multiple classes can be present in the same image. Binary masks ensure that each image contains only one class, making benchmark comparisons more controlled and rigorous.
>
> While benchmarks using binary masks are valuable for research and for some industrial requirements, many industrial applications require fully labeled masks. Through our collaborations with multiple industries, we have observed that models often need to classify all present classes in an image rather than focusing on a single foreground object. To highlight the practical relevance of this approach, we have cited three papers in the introduction that focus on real-world segmentation applications, all of which use fully labeled images. The introduction has been revised to better clarify this point.
>
> >(1-B) The submission made a claim as follows: "Class balance is not assumed, variations are permitted following the distribution of classes within the selected datasets, better reflecting real-world scenarios." The class imbalance is undoubtedly not expected in training distribution, which is well-known as long-tail problem. However, for a benchmark, balance is expected to perform a comprehensive evaluation. While this is more natural, authors seem considering "long-tail" as a good thing for benchmarking, which is quite confusing.
>
> We agree that class imbalance is present in training distributions and that a more balanced class distribution is often preferred for benchmarking. However, in line with the paradigm of semantic segmentation datasets, our goal was to create a benchmark that reflects real-world scenarios. In practical settings, even in an N-way K-shot setup, certain classes naturally appear more frequently than others—for example, in Cityscapes, the class "car" is present in most images. Our intention was not to promote the long-tail problem as desirable but rather to acknowledge its existence and ensure that our benchmark remains representative of real-world data distributions. We modified it in the introduction to emphasize this point.
>
> ### Experiments
>
> >(2-A) Given that Stable Diffusion (SD) is also an important vision foundation models and a few existing studies have explored on using SD to perform few-shot segmentation [1-2], authors are suggested to include related experiments and discussions in their submission.
>
> The papers [1-2] utilize Stable Diffusion either as a technique for augmenting the support set or as a denoising approach to generate masks, which is not directly compatible with our benchmark. The Stable Diffusion model used in [2] is not designed to function as a feature extractor and requires a specialized adaptation process to generate segmentation masks. This makes it fundamentally different from the other models and methods included in our benchmark, preventing a direct comparison.

---

> ### Author Response · Authors · 2025-03-04
> **Response to reviewer B7hM (2/2)**
>
> > (2-B) The proposed setup has its own significance. One aspect is that data like Cityscapes are diverse in object size. For example, a close-up of car compared to a car far away from camera, where instances of car vary in resolution (a.k.a, size). This might be a good perspective to make this paper more solid. Authors are suggested to include such case studies in this paper.
>
> To address this, we conducted an analysis examining the relationship between object size and segmentation performance. Specifically, we plotted the mIoU for the class Car against its pixel area in the image, as shown in the added Figure 7 in the paper.
> From the plot, we observe a strong correlation between object size and segmentation performance. Smaller cars (which are likely farther from the camera) exhibit a higher variance in mIoU and tend to have lower mIoU, while larger cars (closer to the camera) achieve higher mIoU scores. This confirms that object size plays a significant role in few-shot semantic segmentation performance.
>
> However, this improvement in mIoU for larger objects may not necessarily reflect better model generalization. Instead, it could be a consequence of a geometric bias: larger objects have a greater surface area relative to their perimeter, resulting in more inner pixels that are easier to segment accurately. In contrast, smaller objects have a higher proportion of boundary pixels, which are more challenging to segment due to higher edge ambiguity and potential blending with other classes.
>
> We added a section in Appendix A.12 to discuss this effect.
>
> >(2-C) Notice that authors use resolution of 224 for CLIP and MAE, which may make the few-shot segmentation on Cityscapes quite challenging, as the typical input resolution for such data is 1024. This may lead to false conclusion for design choices when performing few-shot segmentation. Authors are suggested to use OpenCLIP ConvNeXT-L, with input resolution of 1120px as presented in this paper [3].
>
> We thank the reviewer for the interesting suggestion. We performed new experiments and found that using a ConvNext-L with a resolution of 1120 does lead to performance improvements; however, it still falls short compared to other vision foundation models (VFMs) such as SAM and DINOv2, which are specifically designed to operate at a resolution of 1024.  In response to this point, we have added a brief discussion in Section 6.3.3 on architecture.
>
> | Model                  | Method  | Cityscapes        | COCO             | PPD              |
> |------------------------|---------|-------------------|------------------|------------------|
> | OpenCLIP (ViT-B)      | Linear  | 21.01 ± 2.28      | 11.64 ± 0.40     | 71.85 ± 9.85     |
> | OpenCLIP (ConvNext-L) | Linear  | 29.29 ± 2.80      | 11.30 ± 0.02     | 81.05 ± 8.30     |
> | OpenCLIP (ViT-B)      | LoRA    | 22.73 ± 1.58      | 14.34 ± 0.06     | 71.55 ± 9.42     |
> | OpenCLIP (ConvNext-L) | LoRA    | 38.52 ± 1.14      | 17.05 ± 0.20     | 84.94 ± 9.72     |

---

### Decision · Action_Editor_sDnZ · 2025-04-05

**Recommendation:** Accept with minor revision

**Comment:**

The paper proposes a new benchmark for the task of few-shot semantic segmentation with foundation models. The reviewers appreciated the writing of the paper, new benchmark, sufficient experimental evaluations and analysis, and also the findings of the paper.  The responses of the authors during the rebuttal period clarified most of the concerns raised by the reviewers. But there are still few important concerns that need to be satisfactorily addressed before the paper can be accepted as is. Specifically, the authors need to satisfactorily justify the claims regarding the full labelling in FSS scenario and also benchmarking using imbalanced data. Also, the authors need to clearly highlight the novelty of the proposed benchmark over the existing benchmarks. The final acceptance of the paper depends on satisfactorily addressing these remaining concerns.

**Audience:**

This paper proposes a new benchmark for the task of few-shot semantic segmentation which is an important research problem of interest to several members in TMLR's audience.

**Claims And Evidence:**

The paper proposes a new benchmark for few-shot semantic segmentation and evaluated 5 foundation models on this. The motivation for creating the new benchmark (as opposed to using the existing ones) is clearly justified. But the novelty of the benchmark with respect to the existing ones should be clearly highlighted. There is still some concerns regarding the other two claims, i.e. full labelling and imbalanced training data, which should be clearly justified or removed. The evaluation of the foundation models and their performance comparison is exhaustive.

---

> ### Author Response · Authors · 2025-04-24
> **Response to Action Editor sDnZ**
>
> We thank the Action Editor for the helpful feedback and the positive recommendation. To clearly highlight the novelty of our benchmark, we have revised the introduction to explicitly articulate how it differs from existing benchmarks such as PASCAL-5ⁱ, COCO-20ⁱ, and GFSS. Our benchmark introduces four key innovations: (1) it avoids training or fine-tuning on a subset of classes before evaluation, focusing instead on direct few-shot adaptation to the target task, closely mirroring practical VFM deployment scenarios; (2) it does not assume disjoint class splits, avoiding unverifiable constraints about class novelty during VFM pretraining; (3) it uses fully labeled support images to enable multi-class supervision, which improves consistency and expressiveness during training; and (4) it preserves the natural class distribution of the dataset, allowing models to be evaluated under more realistic, long-tailed conditions rather than artificially balanced setups.
>
> We have also clarified the motivation behind the use of fully labeled support images and imbalanced data. Binary masks, while useful for controlled experimentation, introduce inconsistencies, where the same class may be labeled as foreground in one episode and background in another, potentially confusing the model during training. Fully labeled support images eliminate this issue and reflect real-world requirements, where models are often expected to segment all visible objects in a scene. Likewise, real-world semantic segmentation datasets (e.g., Cityscapes) inherently exhibit class imbalance, with certain classes appearing far more frequently than others. Rather than masking this challenge, our benchmark retains the natural distribution to better assess model robustness in practical conditions. These points are now clearly stated in the revised introduction to fully address the remaining concerns.